# Deterministic Sparse Fourier Transform for Continuous Signals with Frequency Gap

Xiaoyu Li [1]   Zhao Song [2]   Shenghao Xie [3]

## Abstract

The Fourier transform is a fundamental tool in computer science and signal processing. In particular, when the signal is *sparse* in the frequency domain—having only $k$ distinct frequencies—sparse Fourier transform (SFT) algorithms can recover the signal in a sublinear time (proportional to the sparsity $k$). Most prior research focused on SFT for discrete signals, designing both randomized and deterministic algorithms for one-dimensional and high-dimensional discrete signals. However, SFT for continuous signals (i.e., $x^*(t) = \sum_{j=1}^{k} v_j e^{2\pi \mathbf{i} f_j t}$ for $t \in [0, T]$) is a more challenging task. The discrete SFT algorithms are not directly applicable to continuous signals due to the sparsity blow-up from the discretization. Prior to this work, there is a *randomized* algorithm that achieves an $\ell_2$ recovery guarantee in $\widetilde{O}(k \cdot \mathrm{polylog}(F/\eta))$ time, where $F$ is the bandlimit of the frequencies and $\eta$ is the frequency gap. Nevertheless, whether we can solve this problem without using randomness remains open. In this work, we address this gap and introduce the first sublinear-time *deterministic* sparse Fourier transform algorithm in the continuous setting. Specifically, our algorithm uses $\widetilde{O}(k^2 \cdot \mathrm{polylog}(F/\eta))$ samples and $\widetilde{O}(k^2 \cdot \mathrm{polylog}(F/\eta))$ time to reconstruct the on-grid signal with arbitrary noise that satisfies a mild condition. This is the optimal recovery guarantee that can be achieved by *any* deterministic approach.

## 1. Introduction

The Fourier transform (FT) was introduced by Joseph Fourier in 1822 (Fourier, 1822). Today, it is widely used in computer science and applied mathematics. Its applications include integer multiplication (Fürer, 2009), SUBSET SUM and 3SUM (Cormen et al., 2009; Bringmann, 2017; Koiliaris & Xu, 2017), linear programming (Lee et al., 2019; Jiang et al., 2021), distributional learning (Diakonikolas et al., 2016a;b;c), learning a mixture of regressions (Chen et al., 2020), fast Johnson-Lindenstrauss transform (Lu et al., 2013), and TensorSRHT (Ahle et al., 2020; Song et al., 2021) with its applications to optimization (Song et al., 2024). In applied math, the Fourier transform is a key mathematical tool in solving partial differential equations and performing function approximation (Evans, 2022; Smets et al., 2023; Helwig et al., 2023).

The Fourier transform also has a wide range of real-world applications, including signal processing, electrical engineering, pattern recognition, image/audio/video compression, etc. The famous random Fourier feature methods (Rahimi & Recht, 2007; 2008; Le et al., 2013; Yang et al., 2014; Yu et al., 2016; Tancik et al., 2020; Li et al., 2020; Cheng et al., 2023) bridge the classical Fourier analysis and modern kernel methods. In recent years, Fourier transform has also emerged as a powerful tool within machine learning research, inspiring diverse models and algorithms (Lee et al., 2020; Choromanski et al., 2021; Li et al., 2021; Song & Yu, 2021; Song et al., 2023b; Yu et al., 2023; Yi et al., 2023; Tran et al., 2023; Bonev et al., 2023; Zeng et al., 2024; Tan et al., 2024; Xiao et al., 2024; Chen et al., 2024; Liang et al., 2024; Zhou et al., 2024; 2025; Alman & Song, 2025; Li et al., 2025; Yu et al., 2025). Due to its significance in both theory and practice, finding an efficient algorithm to compute the Fourier transform is of utmost importance.

Since the seminal fast Fourier transform (FFT) algorithm given by Cooley & Tukey (1965), the community has been looking for a faster $o(N \log N)$ time algorithm, but has not yet had success. In contrast, as quoted from Indyk & Kapralov (2014):

> *"Many of these applications rely on the fact that most of the Fourier coefficients of the signals are small or equal to zero, i.e., the signals are (approximately) sparse."*

That is, when the Fourier spectrum is approximately $k$-

[1]University of New South Wales [2]University of California, Berkeley [3]Texas A&M University. Correspondence to: Zhao Song <magic.linuxkde@gmail.com>.

*Proceedings of the 42$^{nd}$ International Conference on Machine Learning*, Vancouver, Canada. PMLR 267, 2025. Copyright 2025 by the author(s).

*sparse* for a *sublinear* parameter $k = o(N)$, we can expect better sublinear-sample/time algorithms. This motivates the studies on the *sparse Fourier transform* problem (Sparse FT).

Over the last two decades, the Sparse FT problem has been extensively studied. The prior works mainly follow two lines: (i) those in the discrete settings (or the compressed sensing literature) (Goldreich & Levin, 1989; Kushilevitz & Mansour, 1993; Mansour, 1995; Gilbert et al., 2002; Akavia et al., 2003; Candes & Tao, 2006; Donoho, 2006; Rudelson & Vershynin, 2008; Blumensath & Davies, 2008; Hassanieh et al., 2012b;a; Cheraghchi et al., 2013; Iwen, 2013; Indyk et al., 2014; Indyk & Kapralov, 2014; Bourgain, 2014; Kapralov, 2016; Haviv & Regev, 2016; Kapralov, 2017; Li & Nakos, 2020; Kapralov et al., 2019; Nakos et al., 2019; Nakos & Song, 2019; Song, 2019); and (ii) those in the continuous setting (Boufounos et al., 2012; Moitra, 2015; Price & Song, 2015; Chen et al., 2016; Chen & Price, 2019b;a; Song, 2019; Jin et al., 2023; Song et al., 2023a).

Although more researchers have concentrated their attention on DFT algorithms, a continuous Fourier transform (CFT) is also essential, since many practical signals are continuous in nature. In the continuous setting, one can observe $x(t) = x^*(t) + g(t)$, where $x^*(t)$ is the ground-truth signal whose frequencies are *real* numbers, and $g(t)$ is a time-varying noise with a suitable signal-to-noise ratio. The heavy frequencies are assumed to have a frequency gap $\eta$, i.e., the minimum distance between two heavy frequencies is at least $\eta$. Moreover, the sparsity parameter $k$ satisfies $k = o(F/\eta)$, where $F$ is the band-limit of the signal. CFT algorithms recover the heavy frequencies by sampling in the time domain.

### 1.1. Problem Formulation

We study the Fourier transform of a continuous $k$-sparse signal defined as follows.

**Definition 1.1** (Continuous-time, $k$-Fourier-sparse signal). *Let $k \in \mathbb{Z}_{>0}$. Let $\delta_{f_i}(f)$ denote the Dirac function centered at $f_i \in \mathbb{R}$. We define the k-Fourier-sparse signal $\widehat{x}^*(f)$ to be as follows:*

$$x^*(t) := \sum_{j=1}^{k} v_j \cdot e^{2\pi \mathbf{i} f_j t} \xrightarrow{\text{CFT}} \widehat{x}^*(f) := \sum_{j=1}^{k} v_j \cdot \delta_{f_j}(f)$$

*where $v_j \in \mathbb{C}$ is the coefficient and $f_j \in \mathcal{F}$ is the frequency contained in the frequency range $\mathcal{F} \subset \mathbb{R}$ for each $j \in [k]$. We use $\mathcal{K}$ to denote the set of $f_j$'s.*

Moreover, we assume that the frequency range $\mathcal{F}$ is a set of equidistant points in $[-F, F]$. See the formal definition below, and we assume that active frequency of signal $x(t)$ can only be taken in a finite set $\mathcal{F}$ defined as below.

**Definition 1.2** (Possible range of active frequency). *For a given frequency gap $\eta$ and bounded range $[-F, F]$, we define the possible range of active frequency as the set*

$$\mathcal{F} := \{i \cdot \eta \mid \forall i \in \mathbb{Z}, \text{ and } i \cdot \eta \in [-F, F]\}.$$

In our observation of signal $x^*(t)$, we receive a time-varying noise, denoted by a continuous function $g(t) : \mathbb{R} \to \mathbb{C}$. The observed $x(t)$ is the sum of the ground-truth signal $x^*(t)$ and $g(t)$. Formally, we define the model of noisy observations, which is commonly used in the literature (e.g. Price & Song (2015); Chen et al. (2016); Song et al. (2023a)).

**Definition 1.3** (Noisy observation). *We define the observed signal $x(t)$ as follows:*

$$x(t) := x^*(t) + g(t) = \sum_{j=1}^{k} v_j e^{2\pi \mathbf{i} f_j t} + g(t),$$

*where $g(t)$ is an arbitrary function.*

Currently, all of the previous sublinear CFT algorithms are randomized. It is natural to define and study the continuous setting where a deterministic algorithm is possible. This work presents a positive answer to that.

### 1.2. Our Result

We state our main result as follows:

**Theorem 1.4** (Main result, informal version of Theorem 3.12). *Let $\mathcal{F} := [-F, -F + \eta, \cdots, -\eta, 0, \eta, \cdots, F - \eta, F]$ denote the candidates of heavy frequency. Let $g(t)$ denote the noise whose Fourier spectrum spans the entire frequency domain, and it satisfies mild assumptions in the time domain. Let $x(t) := x^*(t) + g(t)$ denote the observed signal. For $T \geq \widetilde{\Omega}(1/\eta)$, there exists a deterministic algorithm that observes the signal at time points in $S \subset [0, T]$ of size $|S| = O(k^2 \log k \log^2(F/\eta))$, and recovers all $f_i$ and $v_i$ accurately in $O(k^2 \log k \log^3(F/\eta))$ time.*

We remark that the straightforward approach of discretizing the signal and applying the standard FFT requires at least $F/\eta = \Omega(FT)$ samples to achieve the desired resolution. However, this method is not robust to noise, as it cannot effectively separate the signal components from the noise $g(t)$, which spans the entire frequency domain. In contrast, our algorithm reduces the sample complexity to $\widetilde{O}(k^2)$, which is sublinear in $FT$, by leveraging the sparsity structure of the signal. Furthermore, our method is robust to noise, enabling accurate recovery of frequencies $f_i$ and amplitudes $v_i$ with significantly fewer samples compared to traditional FFT-based approaches.

Moreover, our algorithm achieves nearly optimal sample complexity and runtime due to the $\Omega(k^2)$ deterministic lower bound established in Ganguly (2008); Foucart et al. (2010).

# 2. Preliminaries

In this section, we introduce some basic definitions and tools in sparse Fourier transforms. In Section 2.1, we introduce the notations in this paper. In Section 2.2, we formally define the discrete Fourier transform and the continuous Fourier transform. In Section 2.3, we define the convolution of two functions. In Section 2.4, we introduce the hash functions used in our algorithms. In Section 2.5, we introduce the filter functions that isolate the heavy-hitter in each hashing bucket. In Section 2.6, we define the measurement of the signal under the filter function.

## 2.1. Notations

We use $a \gtrsim b$ to denote $a \geq C \cdot b$ for some constant $C > 0$. Let $n$ be a positive integer, $[n] := \{1, 2, \cdots, n\}$. We use $\mathbf{i} := \sqrt{-1}$, and we use $\omega$ to represent $e^{-2\pi \mathbf{i}}$ for simplicity of notion, e.g., we sometimes write $e^{-2\pi \mathbf{i}}t$ as $\omega^t$. For a complex number $z \in \mathbb{C}$ with $z = a + \mathbf{i}b$ where $a, b \in \mathbb{R}$. We use $|z|$ to denote $\sqrt{a^2 + b^2}$. $z$ can also be expressed as $z = r \cdot e^{\mathbf{i}\theta}$, where $r \in \mathbb{R}_{>0}$ and $\theta \in [0, 2\pi]$. We define $\arg(z) = \theta$. Let $\{z_i\}$ be a sequence of complex numbers. It's median is defined as $\operatorname{median} z_i = \operatorname{median} \operatorname{Re}(z_i) + \mathbf{i} \operatorname{median} \operatorname{Im}(z_i)$. We use $\Pr[]$ to denote probability. We use $\mathbb{E}[]$ to denote expectation. For $x \in \mathbb{R}$, we use $\operatorname{round}(x)$ to denote the integer with the closest distance to $x$. Let $x = i + q$ where $i$ is an integer and $q \in [0, 1)$, we define $x \pmod 1 := q$. Let $x(t) : \mathbb{R} \to \mathbb{C}$ be a function. For a finite set $S \subset \mathbb{R}$, we define $\|x_S\|_1 := \sum_{t \in S} |x(t)|$. Now, let $S$ be a finite sequence, we define $x_S$ as the vector in which the $t$-th entry is $x(S_t)$ where $S_t$ denotes the $t$-th element in $S$. We use $\mathbf{0}_n$ to denote a vector formed by $n$ zeros.

## 2.2. Fourier Transform

We define the discrete Fourier transform (DFT) and the continuous Fourier transform (CFT) below.

**Definition 2.1** (Discrete Fourier transform). *Given a complex vector $x \in \mathbb{C}^n$, we say that $F$ is the discrete Fourier transform matrix if*

$$F_{i,j} := \frac{1}{\sqrt{n}} e^{-2\pi \mathbf{i} \cdot ij/n}.$$

*We define the discrete Fourier transform of $x$ to be*

$$\widehat{x} := Fx.$$

**Definition 2.2** (Continuous Fourier transform). *Given a function $x(t) : [0, T] \in \mathbb{C}$ and $\widehat{x}(f) : [-F, F] \to \mathbb{C}$, the continuous Fourier transform is defined as*

$$\widehat{x}(f) = \int_{-\infty}^{\infty} x(\tau) e^{-2\pi \mathbf{i} f \tau} d\tau,$$

*and the Continuous Inverse Fourier Transform is defined as*

$$x(t) = \int_{-\infty}^{\infty} \widehat{x}(\sigma) e^{2\pi \mathbf{i} \sigma t} d\sigma.$$

## 2.3. Convolution

We define the discrete and the continuous convolution as follows.

**Definition 2.3** (Convolution). *For two functions $f, g$ with same domain $\mathcal{D}$, we have*

$$(f * g)(t) = \int_{\tau \in \mathcal{D}} f(t - \tau) \cdot g(\tau) \tau.$$

*For two vectors $f, g$ with same length $n$, we have*

$$(f * g)[i] = \sum_{f \in [n]} f_{i-j} \cdot g_j.$$

## 2.4. Hash Functions

In this section, we introduce some hash functions used in sparse Fourier transform algorithms.

**Definition 2.4** (Hashing functions, Definition 4.1 in Li & Nakos (2020), Section 3 in Hassanieh et al. (2012b) Definition A.5, A.6, A.7 in Price & Song (2015)). *Let $\sigma \in \mathbb{R}$ and $b \in [-F, F]$. Let $B$ be the number of buckets.*

- *We define function $\pi_{\sigma,b} : \mathcal{F} \to [0, 1]$ to be*

$$\pi_{\sigma,b}(f) := \sigma(f - b) \pmod 1.$$

- *We define function $h_{\sigma,b} : \mathcal{F} \to [B]$ to be*

$$h_{\sigma,b}(f) := \operatorname{round}(B \cdot \pi_{\sigma,b}(f)).$$

- *Fix $f \in \mathcal{F}$, we define function $o_{f,\sigma,b} : \mathcal{F} \to [0, 1]$ to be*

$$o_{f,\sigma,b}(f') := \pi_{\sigma,b}(f') - (1/B)h_{\sigma,b}(f).$$

Specifically, $h_{\sigma,b}(f)$ hashes a frequency $f$ to one of the $B$ buckets, and $o_{f,\sigma,b}(f')$ measures the offset of a frequency $f'$ to the center of the bucket containing $f$.

**Definition 2.5** (Pseudorandom Permutation, Definition 4.2 in Li & Nakos (2020), Definition A.1 in Price & Song (2015)). *For $\sigma \in \mathbb{R}, a \in [0, T], b \in [-F, F]$, the permutation $P_{\sigma,a,b}$ is defined as*

$$(P_{\sigma,a,b}x)(t) = x(\sigma(t - a)) \cdot \omega^{t\sigma b}$$

**Definition 2.6** (Sequence of Hashings, Definition 4.4 in (Li & Nakos, 2020)). *We use $\{(\sigma_r, a_r, b_r)\}_{r \in [d]}$ to denote the parameters of a sequence of $d$ hashings. Each $(\sigma_r, a_r, b_r)$ specifies three hashing functions: $\pi_{\sigma_r,b_r}, h_{\sigma_r,b_r}, o_{f,\sigma_r,b_r}$, and one pseudo permutation: $P_{\sigma_r,a_r,b_r}$.*

**Definition 2.7** (Tuple of Hashing). *We use $H = (\sigma, a, b)$ to denote a tuple of hashing. In a sequence of hashings $\{(\sigma_r, a_r, b_r)\}_{r \in [d]}$, we use $H_r$ to represent $(\sigma_r, a_r, b_r)$.*

## 2.5. Filter Functions

Filtering is one of the most important techniques for sparse Fourier transform, which allows us to isolate each individual frequency and reduces the $k$-sparse signal to a set of "almost" one-sparse signals.

**Definition 2.8** (Filter function in the continuous setting, Definition A.3 in Price & Song (2015), see discrete variations in Definition 4.6 in Li & Nakos (2020), Definition 2.3 in Kapralov (2017), Definition 2.3 in Hassanieh et al. (2012b)). *Let $B \in \mathbb{Z}_{>0}$ be a power of two. Let $N$ be some fixed integer. Let offset $o$ be defined as Definition 2.4. We say $\widehat{G} : [0,1] \to \mathbb{R}$, with $G$ being its Fourier transform, is a flat filter with $B$ buckets, sharpness $\epsilon$ if the followings hold:*

- *Property 1: $\widehat{G}_{o_{f,\sigma_r,b_r}} \in [0,1]$ for all $o_{f,\sigma_r,b_r} \in [0,1]$*

- *Property 2: $\widehat{G}_{o_{f,\sigma,b}} \geq 1 - \epsilon$ for all $o_{f,\sigma,b} \in [-\frac{1}{2B}, \frac{1}{2B}]$*

- *Property 3: $\widehat{G}_{o_{f,\sigma,b}} \leq \epsilon$ for all $o_{f,\sigma,b} \in [0,1] \backslash [-\frac{1}{B}, \frac{1}{B}]$*

- *Property 4: $\sum_{i \in \mathbb{Z}} G(i)^2 = O(\frac{1}{B})$*

- *Property 5: $\operatorname{supp}(G) \subset [-D, D]$ where $D$ is $O(\log(B))$ rounding to the closest integer*

**Remark 2.9.** *The construction of $(G, \widehat{G})$ can be found in (Price & Song, 2015) and (Jin et al., 2023). Notice that $\widehat{G} : [0, 2\pi] \to \mathbb{R}$ in their construction, we can simply extend to our setting by scaling.*

## 2.6. Measurement

This section defines the notion of measurement. It formalizes the output of a central subroutine HASHTOBINS presented later, which recovers the active tones by performing DFT on the filtered signals in each hashing bucket. The following definition is the measurement without noise.

**Definition 2.10** (Measurement without noise, implicitly in Lemma 3.4 in Price & Song (2015), see discrete variation in Definition 4.8 in in Li & Nakos (2020)). *Let the signal and frequencies $\widehat{x}, v$ be defined as Definition 1.1. Let $H = (\sigma, a, b)$ be a tuple of hashing. Let $\widehat{G}$ be a flat filter with $B$ buckets and sharpness $\epsilon$ (refer to Definition 2.8). A measurement $m_H(\widehat{x}(f)) \in \mathbb{C}^B$ is defined as, for all $h_{\sigma,b}(f) \in [B]$,*

$$(m_H(\widehat{x}(f)))_{h_{\sigma,b}(f)} = \sum_{f \in \mathcal{F}} \widehat{G}_{o_{f,\sigma,b}(f')} \cdot \omega^{a\sigma f} \cdot v_f,$$

*where $\pi$ is a hash function induced from $H$.*

The next statement states an equivalent formulation of the measurement.

**Claim 2.11.** *Under the conditions of Definition 2.10, we have*

$$(\widehat{G}_{o_{f,\sigma,b}(f)})^{-1} \cdot (m_H(\widehat{x}(f)))_{h_{\sigma,b}(f)} \cdot \omega^{-a\sigma f}$$
$$= v_f + (\widehat{G}_{o_{f,\sigma,b}(f)})^{-1} \cdot \sum_{f' \in \mathcal{F} \backslash \{f\}} \widehat{G}_{o_{f,\sigma,b}(f')} \cdot v_{f'} \cdot \omega^{a\sigma(f'-f)}.$$

*Proof.* By the definition of $m_H$ (Definition 2.10), we have

$$(\widehat{G}_{o_{f,\sigma,b}(f)})^{-1} \cdot (m_H)_{h_{\sigma,b}(f)} \cdot \omega^{-a\sigma f}$$
$$= (\widehat{G}_{o_{f,\sigma,b}(f)})^{-1} \cdot \Big( \sum_{f' \in \mathcal{F}} \widehat{G}_{o_{f,\sigma,b}(f')} \cdot \omega^{a\sigma f'} \cdot v_{f'} \Big) \cdot \omega^{-a\sigma f}$$
$$= v_f + (\widehat{G}_{o_{f,\sigma,b}(f)})^{-1} \cdot \sum_{f' \in \mathcal{F} \backslash \{f\}} \widehat{G}_{o_{f,\sigma,b}(f')} v_{f'} \omega^{a\sigma(f'-f)}$$

Thus the proof is complete. $\square$

## 3. Technical Overview

In this section, we provide an overview of our contribution. We start by summarizing the framework in (Li & Nakos, 2020) in Section 3.1, then present our algorithm's motivation and details in Section 3.2.

## 3.1. Summary of previous works

Sparse FFT algorithm searches for the active frequency by binning them into a small number of bins. In the discrete setting, Hassanieh et al. (2012b;a) introduced new methods for locating the isolated signal and updating the signal by directly filtering the bins, which improved the time and sample complexity. Indyk & Kapralov (2014) presented a recursive single-entry reduction algorithm which gives a $\ell_\infty$ norm guarantee. Based on their result and a modified HASHTOBINS with initial guess from Kapralov (2017), Li & Nakos (2020) introduced a deterministic algorithm by de-randomization w.r.t. the hashing functions. In the continuous setting, Price & Song (2015); Chen et al. (2016) defined the k-sparse continuous signal and used a randomized time-sampling technique to control time-varying noise. They extended the guarantee of the fast DFT algorithm to the continuous setting by identifying between CFT, DTFT, and DFT, which is helpful in the analysis of our work. Jin et al. (2023) is a higher-dimensional generalization of Price & Song (2015). In this work, we provide the first deterministic continuous sparse FT algorithm. We result in a $\ell_1/\ell_2$-mixed norm guarantee for error. We also note that another line of work focuses on the sparse Fourier transform over Boolean hypercube or Abelian groups (e.g., Goldreich & Levin (1989); Akavia et al. (2003); Iwen (2007); Akavia (2010)). They have important applications in Boolean function analysis and complexity theory. However, the settings of these papers are very different from the compressed sensing fashion and thus beyond the scope of our work.

**Contribution of Li & Nakos (2020) and their limitation in the continuous setting.** During the hashing and detection of active signal, it is possible that two distinct active frequencies are hashed into the same bin and hence cannot be recovered. Li & Nakos (2020) gave a formal definition of this event and used a de-randomization strategy to find a fixed sequence of hashing functions that prevent this event. Their method uses a pessimistic estimator to upper-bound the possibility of bad events and reduce it by choosing proper hashing parameters. Then, they embedded this deterministic HASHTOBINS algorithm into the $\ell_\infty$ norm reduction algorithm in Indyk & Kapralov (2014) to reach the final result.

Notice that the strategy of Li & Nakos (2020) is not directly feasible for continuous signals. Because there is only a finite light-hitter in the discrete setting, it can traverse all frequency points and generate a deterministic hashing sequence. However, we need to deal with the continuous noise function in our algorithm. Moreover, the hashing scheme in the discrete setting cannot be applied in the continuous setting. Unlike DFT, our CFT algorithm takes samples from the time interval of unequal length with the active frequency set, leading to a different hashing and filtering strategy. We must change our de-randomization steps to fit the new hashing functions.

### 3.2. Our techniques

In this work, we generalize Li & Nakos (2020) to the continuous setting and overcome the limitations. In this section, we first discuss the problem setting. Then, we show how to generate the de-randomized hashing sequence under the hashing scheme of sparse CFT. Next, we propose a reasonable noise model ($(C, \xi)$-noise, Definition 3.3) that enables an efficient and robust deterministic sparse Fourier transform algorithm. We further show how to combine the continuous HASHTOBINS with the de-randomized hash sequence. Then we incorporate it with a recursive sparse recovery algorithm, which leads to our main theorem (Theorem 3.12).

In Section 3.2.1, we summarize our de-randomization steps. In Section 3.2.2, we define the $(C, \xi)$-noise. In Section 3.2.3, we introduce the continuous variate of an essential subroutine HASHTOBINS. In Section 3.2.4, we provide our main algorithm.

### 3.2.1. DE-RANDOMIZATION.

To discover the $k$ active frequencies from $\mathcal{F}$, we use a hashing and filtering method. First, we hash the points in $\mathcal{F}$ into $B = O(k)$ buckets (where each bucket is the union of comb-like, equispaced intervals on the real line). Then, we use a pair of filter functions $(G, \widehat{G})$ to select the active frequencies. $\widehat{G}$ is constructed to be close to 1 in the center of its domain and close to 0 elsewhere. Let $\sigma, b$ be the pa-

rameters of the hash function. We use function $o_{f,\sigma,b}(f')$ to measure the distance from the hashing of $f'$ to the center of the bucket[1] where $f$ is hashed into. (See Definition 2.4 for the formal definitions of the hash functions.) Morally, if $o_{f,\sigma,b}(f)$ is small and $o_{f,\sigma,b}(f')$ is big, i.e., $f$ is hashed close to the center of a bucket, while $f'$ is not close or in a different bucket, then we can discover $f$ by the filter function. However, two active frequencies, $f$ and $f'$, may hash to the same bucket, obstructing the discovery. This motivates us to define bad events as follows. Intuitively, it says that although we cannot guarantee that the collision of $f$ and $f'$ does not happen in a single turn of hashing, we can control the total time of the collision in a sequence of hashing. Then, we can run the algorithm multiple times and take the median of outputs to reach a good approximation.

**Definition 3.1** (Bad Events $A_{f,f'}$). *We use $d$ to denote the time we call* HASHTOBINS *in one round of sparse detection. We use $\beta$ as a factor depending on $|\mathcal{F}|$, which will be determined later. Let $\{(\sigma_r, b_r)\}_{r=1}^d$ be a sequence of hashing parameters. For any $f, f' \in \mathcal{F}$, $f \neq f'$, we define $A_{f,f'}$ to be the event that*

$$\sum_{r=1}^{d} \widehat{G}_{o_{f,\sigma_r,b_r}(f')} \geq \beta.$$

Previous CFT research (e.g. Price & Song (2015); Jin et al. (2023)) used a randomized hashing function to control the impact of bad events. They bounded the expectation of error in each stage and repeated multiple stages to reach a small failure probability. Different from them, this work finds a sequence of deterministic hashing function $\{(\sigma_r, b_r)\}_{r=1}^d$, which prevents the happening of bad events. We use $h_r$ to denote a pessimistic estimator, tracking the probability of undesirable events (such as hash collisions) given the first $r$ selected hash functions. The recursive procedure can be summarised as follows.

1) Initial state: Let $h_r(f, f', \sigma_1, b_1, \cdots, \sigma_r, b_r)$ be a sequence of function satisfying

$$\sum_{f, f' \in \mathcal{F}: f \neq f'} h_0(f, f') < 1$$

Notice that $h_0(f, f')$ is determined at initial state while $h_r(f, f')$ depends on $\sigma_r, b_r$ which are chosen later.

2) De-randomization step: Let $\Pr[A_{f,f'} \mid \sigma_1, b_1, \ldots, \sigma_r, b_r]$ be defined as the probability of bad events conditioned on $\{(\sigma_k, b_k)\}_{k=1}^r$. Given $h_r(f, f', \sigma_1, b_1, \cdots, \sigma_r, b_r)$, we choose $\sigma_{r+1}, b_{r+1}$ to satisfy the inequalities below.

$$h_{r+1}(f, f'; \sigma_1, b_1, \cdots, \sigma_{r+1}, b_{r+1})$$
$$\geq \Pr[A_{f,f'} \mid \sigma_1, b_1, \ldots, \sigma_{r+1}, b_{r+1}],$$

---

[1]More precisely, the center of any internal that contains $f'$ in that bucket.

and

$$h_{r+1}(f, f'; \sigma_1, b_1, \cdots, \sigma_{r+1}, b_{r+1})$$
$$\leq h_r(f, f', \sigma_1, b_1, \cdots, \sigma_r, b_r).$$

3) Final state: The procedure ends at $r = d$.

This process outputs $\{(\sigma_r, b_r)\}_{r=1}^d$ such that

$$\sum_{f, f' \in \mathcal{F}: f \neq f'} \Pr[A_{f,f'} \mid \sigma_1, b_1, \ldots, \sigma_d, b_d] < 1.$$

Since $A_{f,f'} \mid \sigma_1, b_1, \ldots, \sigma_d, b_d$ is a determined event, the probability of the occurrence of bad events is zero. Using this specific hashing tuple sequence, we can safely hash the possible active frequency points. We note that this process does not depend on the observed signal, and the good hashing parameters can be found efficiently in the preprocessing (see Definition A.6 for a more detailed discussion on the pessimistic estimator).

**Lemma 3.2** (De-randomization, Informal Version of Lemma A.21). *Let $\mathcal{F}$ be the range of active frequency defined in Definition 1.2. Let $C_1$ be some fixed constant in $(\frac{1}{2}, 1)$. Let $B \in \mathbb{Z}_{>0}$ be a power of 2. Let $\epsilon := \frac{20}{B}$, $\beta := \frac{6}{C_1} \cdot \log |\mathcal{F}|$, and $d := \frac{3C_1}{40} \cdot B \log |\mathcal{F}|$. Let $H_d = \{(\sigma_r, a_r, b_r)\}_{r \in [d]}$ be a sequence of hashing chose by procedure in Definition A.6. Let $\widehat{G}$ be a flat filter in accordance of hashing functions in $H_r$ (see Definition 2.8). Then it holds that, for all $f, f' \in \mathcal{F}$ with $f \neq f'$,*

$$\sum_{r \in [d]} \widehat{G}_{o_{f,\sigma_r,b_r}}^{-1}(f) \widehat{G}_{o_{f,\sigma_r,b_r}}(f') \leq \frac{\beta}{1 - \epsilon}.$$

### 3.2.2. $(C, \xi)$-NOISE

Similar to controlling the effect of bad events, Price & Song (2015); Jin et al. (2023) use the randomness of their sample technique to de-noise. However, it is hard to tackle the noise when we apply a deterministic algorithm, since we can only take samples at finite points. For example, if $g(t)$ is extremely large at our fixed sample points compared to its integral, then it will disturb our observation and prevent the active frequency from being detected. Therefore, we need to introduce extra assumptions on the upper bound of the noise function. Our assumption consists of the energy bound and a $g(t)$-dependent factor $\xi$, which describes the suitability of $g(t)$ for a deterministic algorithm. Formally, we define the $(C, \xi)$-noise as follows.

**Definition 3.3** ($(C, \xi)$-noise). *Let $g(t) : [0, T] \to \mathbb{R}$ be the noise function. Let $C > 0$ be some fixed constant. Let $\xi$ be a parameter depending on $g(t)$. Then we say $g(t)$ is a $(C, \xi)$-noise if it satisfies*

$$\max_{t \in [0,T]} g(t)^2 \leq C \cdot \frac{1}{T} \int_0^T g(t)^2 \mathrm{d}t + \xi.$$

A canonical example satisfying the $(C, \xi)$-noise condition is any bi-Lipshitz function. A function $g : [0, T] \to \mathbb{R}$ is $(L_1, L_2)$-bi-Lipschitz if for all $t_1, t_2 \in [0, T]$, it holds that

$$L_2 |t_1 - t_2| \leq |g(t_1) - g(t_2)| \leq L_1 |t_1 - t_2|.$$

The next statement shows how it satisfies our assumption.

**Lemma 3.4.** *Let $g$ be a integrable $(L_1, L_2)$-bi-Lipschitz function. Then it satisfies the condition of the $(C, \xi)$-noise.*

*Proof.* Let $g_{\min}$ minimum value of $g(t)$ and let $t^*$ be the time that achieves the minimum. Then we have

$$\int_0^T g(t)^2 \mathrm{d}t - T \cdot g_{\min}^2$$
$$= \int_0^T g(t)^2 - g_{\min}^2 \mathrm{d}t$$
$$= \int_0^T (g(t) + g_{\min})(g(t) - g_{\min}) \mathrm{d}t$$
$$\geq \int_0^T g_{\min} \cdot L_1 |t - t^*| + L_1 |t - t^*|^2 \mathrm{d}t$$
$$\geq C(L_1) * (g_{\min} T^2 + T^3),$$

where the third step follows from the definition of the $(L_1, L_2)$-bi-Lipschitz function, i.e., $L_2 |x_1 - x_2| \leq |g(x_1) - g(x_2)|$, and the last step hides constant $C(L_1)$ that depends on $L_1$. On the other hand, let $g_{\max}$ be the maximum value of $g(t)$, we have $g_{\max} \leq g_{\min} + L_2 T$, therefore,

$$g_{\max}^2 \leq C(L_1, L_2) \cdot \frac{1}{T} \int_0^T g(t)^2 \mathrm{d}t,$$

where $C(L_1, L_2)$ is a constant that depends on $L_1$ and $L_2$. Thus, $g(t)$ satisfies the definition of the $(C, \xi)$-noise by taking $G = C(L_1, L_2)$ and $\xi = 0$. $\square$

Therefore, our assumption is mild and suitable for a wide variation of noise functions.

### 3.2.3. DETERMINISTIC HASHTOBINS UNDER CONTINUOUS SETTING

HASHTOBINS is a commonly-used algorithm in FFT. It takes discrete samples from the time interval and uses DFT to measure the signal in each hashing bucket. This measurement reflects the tone of active frequency in the corresponding bucket. Kapralov (2017) applies a modified HASHTOBINS algorithm to determine the active tones in the discrete setting recursively. Our work extends this algorithm to the continuous setting.

For simplicity of notation, we define a vector $v_f \in \mathbb{C}^{|\mathcal{F}|}$ as below to represent the k-sparse tones of signal $\widehat{x}^*(f)$.

**Algorithm 1** HashToBins

1: **procedure** HASHTOBINS($x, \widehat{z}, H = (\sigma, a, b)$)
2:                                                     ▷ Lemma 3.6
3:     **for** $j \in [BD]$ **do**
4:         $y_j \leftarrow G(j) \cdot P_{\sigma,a,b}(x)(j)$
5:     **end for**
6:     **for** $j \in [B]$ **do**
7:         $u_j \leftarrow \sum_{j\in[D]} y_{Bi+j} - \widehat{G} * \widehat{P_{\sigma,a,b}(z)}(j/B)$
8:     **end for**
9:     **return** The DFT $\widehat{u} \in \mathbb{C}^B$ of $u$
10: **end procedure**

**Definition 3.5.** *Consider $\{f_i\}_{i=1}^{|\mathcal{F}|}, f_i \in \mathcal{F}$ as a finite sequence of points ordered by their magnitude in the frequency interval. Recall $\widehat{x}^*(f) = \sum_{j=1}^{k} v_j \cdot \delta_{f_j}(f)$. Let $v_f := v_i$ if $f_i$ is an active frequency; otherwise, $v_f := 0$.*

Given an initial guess $\widehat{z} \in \mathbb{C}^{|\mathcal{F}|}$ and a discrete sample of signal $x(t)$, this algorithm can return the bucket-wise measurement of the difference between $v$ and $\widehat{z}$. The guarantee of this procedure is stated as follows.

**Lemma 3.6** (HASHTOBINS, Informal Version of Lemma B.5)**.** *We use $B$ to denote the number of buckets. We use $h_{\sigma,b}(f)$ to denote the index of the bucket where $f$ is hashed into. Given a vector $\widehat{z} \in \mathbb{C}^{|\mathcal{F}|}$, there exists a deterministic procedure HASHTOBINS which computes $u \in \mathbb{C}^B$ with the following guarantee: for any $f \in \mathcal{F}$,*

$$\left| u_{h_{\sigma,b}(f)} - \sum_{f'\in\mathcal{F}} \widehat{G}_{o_{f,\sigma,b}(f')}(v_{f'} - \widehat{z}_{f'})\omega^{a\sigma f'} \right|$$
$$\leq O\left( \frac{\log k}{k} \cdot \left( \frac{C}{T} \int_0^T |g(t)|^2 \mathrm{d}t + \xi \right)^{\frac{1}{2}} \right),$$

*The algorithm takes $O(B\log(B))$ samples. The time complexity of the algorithm is $O(B\log^2(B) + B \cdot \log(F/\eta))$.*

The analysis of this lemma combines the bound for noiseless input (HASHTOBINS($x^*(t), \widehat{z}_f$)) and noise-only input (HASHTOBINS($g(t), \mathbf{0}_{|\mathcal{F}|}$)), where the former has a similar performance to the discrete setting, and the latter is controlled by our assumption in $(C,\xi)$-noise. Hence, our upper bound on error is the sum of $\|\widehat{z}\|_2 \cdot k^{-c}$ and the energy-$\xi$ bound.

As mentioned, the de-randomization step finds a deterministic sequence of hashing parameters that avoids bad events. Taking the median of output of HASHTOBINS($x, \widehat{z}, (\sigma_r, b_r)$) for $r \in [d]$ gives a close approximation of $v_f - \widehat{z}_f$.

**Lemma 3.7** (HASHTOBINS with De-randomized hash sequence, informal version of Lemma B.7)**.** *Let $B := \Theta(k)$ be a power of 2. Let $\widehat{w}_f := v_f - \widehat{z}_f$. Let $\{H_r\}_{r\in[d]} = (\sigma_r, b_r)$ be the sequence of hashing found*

*by the de-randomization process. Let $u_r$ be the output of HASHTOBINS($x, \widehat{z}, H_r$). We define*

$$\mathcal{N}(\widehat{w})$$
$$:= \frac{1}{\alpha k} \sum_{f\in\mathcal{F}} |\widehat{w}(f)| + O\left( \frac{\log k}{k} \left( \frac{C}{T} \int_0^T |g(t)|^2 \mathrm{d}t + \xi \right)^{\frac{1}{2}} \right).$$

*Then, we have, for all $f \in \mathcal{F}$,*

$$\left| \widehat{w}_f - \operatorname*{median}_{r\in[d]} \widehat{G}_{o_{f,\sigma_r,b_r}(f)}^{-1}(u_r)_{h_{\sigma_r,b_r}(f)} \right| \leq \mathcal{N}(\widehat{w}).$$

#### 3.2.4. RECURSIVE SPARSE RECOVERY ALGORITHMS

Finally, we construct our main algorithm by embedding HASHTOBINS into a recursive sparse recovery algorithm, which is in line with the idea of Indyk & Kapralov (2014) and (Li & Nakos, 2020). In each iteration, we run HASHTOBINS to measure the difference between the true tone $v_f$ and the approximated tone $\widehat{z}_f$. Then, we use a threshold $\nu$ to determine whether to change the approximated tone. We set the recovery threshold $\nu = O(\mathcal{N})$ in the initial stage and scaled by a constant $\gamma$ in each iteration. This gives us a super-linear time sparse recovery algorithm (see Appendix B). Since detection is applied to each entry of $v$, this algorithm has an error bound in $\ell_\infty$ norm.

Now, we discuss our sublinear algorithm. In each iteration, we run HASHTOBINS $O(\log(F/\eta))$ times to estimate $v_f \cdot \omega^{\mathbf{i}q\theta_f}$ for each heavy-hitter $f$. Here, $q \in Q := \{2^0, 2^1, \cdots, 2^{\log n - 1}\}$ is the scaling factor we apply on the phase of each heavy-hitter. Then, we adopt a discrete one-sparse recovery technique in Li & Nakos (2020) to recover each $f$. The algorithm is displayed in Algorithm 2.

**Algorithm 2** One Sparse Recovery

1: **procedure** ONESPARSERECOVERY($x_Q$)
2:                                                    ▷ Lemma 3.8
3:     **for** $q \in Q$ **do**
4:         $d_l \leftarrow 2l\pi + \arg(x_q/x_0)$
5:         $I_q \leftarrow \bigcup_{l=0}^{q-1} [\frac{d_l}{q} - \frac{\pi}{4q}, \frac{d_l}{q} + \frac{\pi}{4q}]$
6:     **end for**
7:     $S_0 \leftarrow I_1$
8:     **for** $r \in \{1, 2, \cdots, \log|\mathcal{F}| - 1\}$ **do**
9:         $S_{r+1} \leftarrow S_r \cap I_{2^{r+1}}$
10:     **end for**
11:     Find $\theta_{f'}$ from $S_{\log|\mathcal{F}|-1}$ where $f' \in \mathcal{F}$
12:     $f' \leftarrow \theta_{f'} \cdot \frac{|\mathcal{F}|}{2\pi}$
13:     **return** $f'$
14: **end procedure**

We next state the guarantee of Algorithm 2.

**Lemma 3.8** (One-Sparse Recovery, Lemma 6.1 in Li & Nakos (2020))**.** *Suppose that $|\mathcal{F}|$ is a power of 2. Let $Q :=$*

$\{0, 2^0, 2^1, 2^2 \cdots, |\mathcal{F}|/2\}$. *Let $x \in \mathbb{C}^{\mathcal{F}}$ with the discrete Fourier transform $\widehat{x}$. Let $x_f$ be the $f$-th entry of $x$. Let $\theta_f := \frac{2\pi}{|\mathcal{F}|} f' \mod 2\pi$. Let $\{x_q\}_{q \in Q}$ be a sequence of metric of $x_f$ satisfying*

$$|\arg(x_q) - (\arg \widehat{x}_f + q\theta_f)| \le \pi/8$$

*Then* ONESPARSERECOVERY*(Algorithm 2) recovers $f$ by $\{x_q\}_{q \in Q}$ in $O(\log F/\eta)$ time.*

Instead of traversing through each possible frequency as in the super-linear time algorithm, we locate the heavy-hitters first, and then we do the recursive estimation of the tone of each heavy-hitter. This reduces the $O(F/\eta)$ time recovery to an $O(k)$ time recovery, leading to the sublinear result. Algorithm 3 embeds the above one-sparse recovery procedure to recover the signal in sublinear time.

---

**Algorithm 3** Sublinear-time Sparse Recovery for $\widehat{x} - \widehat{z}$

1: **procedure** SUBLINEAR$(x, \widehat{z}, \nu)$     ▷ Lemma 3.10
2:     $S \leftarrow \emptyset$
3:     **for** $r = 1 \to d$ **do**
4:         **for** $q \in Q$ **do**
5:             $u_q \leftarrow$ HASHTOBINS$(x, \widehat{z}, (\sigma_r, q, b_r))$
6:                 ▷ Lemma 3.6
7:         **end for**
8:         **for** $b = 1 \to B$ **do**
9:             $f \leftarrow$ ONESPARSERECOVERY$(\{(u_q)_b\}_{q \in Q})$
10:                 ▷ Lemma 3.8
11:             $S \leftarrow S \cup \{f\}$
12:             $v_{f,r} \leftarrow \widehat{G}^{-1}_{o_{f,\sigma_r,b_r}(f)}(u_0)_{h_r(f)}$
13:         **end for**
14:     **end for**
15:     $\widehat{w}'_f \leftarrow 0$
16:     **for** $f \in S$ **do**
17:         $v_f \leftarrow \text{median}_{r \in [d]} v_{f,r}$     ▷ Lemma 3.9
18:         **if** $|v_f| > \nu/2$ **then**
19:             $\widehat{w}' \leftarrow v_f$
20:         **end if**
21:     **end for**
22:     **return** $\widehat{w}'$
23: **end procedure**

---

Next, we provide the guarantee of the sublinear-time sparse recovery for $\widetilde{x} - \widetilde{z}$. With the guarantee of de-randomization (Lemma 3.2), the following lemma shows how well the hashing sequence separates different heavy indices.

**Lemma 3.9** (Separating Heavy Indices, informal version of Lemma C.3). *Let $H_r = \{\sigma_r, a_r, b_r\}_{r \in d}$ be a sequence of hashing defined in Definition 2.6. We have $\widehat{G}$ being a flat filter with $\epsilon$ buckets and sharpness $\epsilon$ (see Definition 2.8). Let $\widehat{x}, v$ be defined as Definition 1.1. If for all $f, f' \in \mathcal{F}$ with*

*$f \neq f'$ it holds that,*

$$\sum_{r \in [d]} \widehat{G}^{-1}_{o_{f,\sigma_r,b_r}}(f) \widehat{G}_{o_{f,\sigma_r,b_r}}(f') \le \frac{\beta}{1 - \epsilon}$$

*where $\beta$ is chosen the same as Lemma 3.2, then for any $f$, at least $0.8d$ indices $r \in [d]$ satisfy*

$$|\sum_{f' \in \mathcal{F} \backslash \{f\}} \widehat{G}_{o_{f,\sigma_r,b_r}}(f') \widehat{x}_f| \le \Theta(\frac{1}{B}) \cdot \sum_{f' \in \mathcal{F} \backslash \{f\}} |v_{f'}|.$$

Next, once we have "good" hash sketches (Lemma 3.9), and provided we choose $\nu$ large enough, Algorithm 3 recovers all large entries and approximates them well.

**Lemma 3.10** (Sublinear-time sparse recovery for $\widetilde{x} - \widetilde{z}$, informal version of Lemma C.4). *Let $\widehat{x}, v$ be defined as Definition 1.1. Let $\widehat{z} \in \mathbb{C}^{|\mathcal{F}|}$. Let $B$ be chosen as Definition B.6. Let $\widehat{w}_f := v_f - \widehat{z}_f$. Let $\mathcal{N}(\widehat{w})$ be defined as in Lemma 3.7. Let $\nu \ge 16\mathcal{N}(\widehat{w})$ be a constant to denote a threshold for heavy index. Then the output of the Procedure* SUBRECOVERY *(Algorithm 3) $\widehat{w}'$ satisfies:*

- $|\widehat{w}_f| \ge (7/16)\nu$ *for all $f \in \text{supp}(\widehat{w}')$,*

- $|\widehat{w}_f - \widehat{w}'_f| \le |\widehat{w}_f|/7$ *for all $f \in \text{supp}(\widehat{w}')$,*

- $\{f \in \mathcal{F} : |\widehat{w}_f| \ge \nu\} \subseteq \text{supp}(\widehat{w}')$.

Finally, similar as Hassanieh et al. (2012a) and Li & Nakos (2020), we introduce the definition of signal-to-noise ratio $R^*$. It measures the ratio between each tone's magnitude and the average noise. We assume $R^* = O(\text{poly}(F/\eta))$, which allows us to run only $O(\log(F/\eta))$ iterations of one-stage sparse recovery.

**Definition 3.11** (Signal-to-Noise Ratio). *We define the average of noise $\nu$ as*

$$\mu := O\left(C \cdot \frac{1}{kT} \int_0^T |g(t)|^2 \mathrm{d}t + \xi\right).$$

*Then, the signal-to-noise ratio $R^*$ is defined as*

$$R^* := \|v\|_\infty/\mu.$$

Based on all the above discussions, we obtain the main theorem. Here, we only present the sublinear time result since it outperforms the other algorithm when $F/\eta \gg k$. Our main algorithm is displayed in Algorithm 4. It calls the SUBLINEAR procedure iteratively with geometrically decreasing threshold $\nu^{(t)}$, which recovers the sparse signal in $\log_\gamma R^*$ rounds.

The next statement shows our main result.

**Algorithm 4** Sublinear-time sparse recovery for $\widehat{x}$

1: **procedure** MAIN($x \in \mathbb{C}^n$)       ▷ Theorem 3.12
2:      $T^* \leftarrow \log_\gamma R^*$
3:      $\widehat{z}^{(0)} \leftarrow \mathbf{0}_{|\mathcal{F}|/\eta}$
4:      $\nu^{(0)} \leftarrow C\mu\gamma^T$
5:      **for** $t = 0 \to T^* - 1$ **do**
6:         $\widehat{z}^{(t+1)} \leftarrow \widehat{z}^{(t)} + \text{SUBLINEAR}(x, \widehat{z}^{(t)}, \nu^{(t)})$   ▷
     Algorithm 3
7:         $\nu^{(t+1)} \leftarrow \nu^{(t)}/\gamma$
8:      **end for**
9:      **return** $\widehat{z}$
10: **end procedure**

---

**Theorem 3.12** (Main result, formal version of Theorem 3.12). *Consider any k-Fourier-sparse signal $x^*(t) = \sum_{j=1}^k v_j \cdot e^{2\pi \mathbf{i} f_j t}$ with on-grid frequencies $f_j \in \mathcal{F}$ for band-limit $F$ and gap $\eta$ (Definitions 1.1 and 1.2). Let $T \geq \widetilde{\Omega}(1/\eta)$. For $t \in [0,T]$, let $x(t) = x^*(t) + g(t)$ be the noisy observation with signal-to-noise ratio $R^* = \text{poly}(F/\eta)$, where $g(t)$ is a $(C, \xi)$-noise (Definition 3.3) for some $C, \xi > 0$. We define the noise level as*

$$\mathcal{N} := \frac{1}{k} \sum_{f \in \mathcal{F}} |v_f| + \frac{\log k}{k} \cdot \left( \frac{C}{T} \int_0^T |g(t)|^2 \mathrm{d}t + \xi \right)^{\frac{1}{2}}$$

*Then, there exists a* deterministic *algorithm (Algorithm 4) which finds an $O(k)$-sparse vector $\widehat{z} \in \mathbb{C}^{|\mathcal{F}|}$ such that*

$$|v_j - \widehat{z}_{f_j}| \leq O(\mathcal{N}) \quad \forall j \in [k], \quad and$$
$$|\widehat{z}_f| \leq O(\mathcal{N}) \quad \forall f \notin \{f_1, \ldots, f_k\}.$$

*The algorithm takes $O(k^2 \log k \log^2(F/\eta))$ samples and runs in $O(k^2 \log k \log^3(F/\eta))$ time.*

*Proof.* **Proof of Correctness.**

Notice that we have the same guarantees in Lemma 3.10 as in Lemma B.8. Therefore, in Algorithm 4, we directly replace the super-linear time sparse recovery algorithm SUPERLINEAR with the sublinear time sparse recovery algorithm SUBLINEAR. Then, the proof of the correctness is the same as that of super-linear time algorithm (Algorithm 6 in Appendix B).

**Proof of Sample Complexity.**

The sample complexity of the algorithm is counted as below,

Sample Complexity
$= T^* \cdot \text{SUBLINEAR}$
$= \log(F/\eta) \cdot \text{SUBLINEAR}$
$= \log(F/\eta) \cdot d \cdot \text{HASHTOBINS}$
$= \log(F/\eta) \cdot d \cdot O(B \log(B))$

$= \log(F/\eta) \cdot O(B \log(F/\eta)) \cdot O(\log B) \cdot O(B)$
$= O(k^2 \log k \cdot \log^2(F/\eta))$

where the first step is because we run $T^*$ times SUBLINEAR, the 2nd step holds since $T^* = O(\log(F/\eta))$, the 3rd step is because we have $d$ hashing tuples (notice that the samples for $u_q$ are reused in each iteration, so we only run $d$ times HASHTOBINS), the 4th step is by Lemma 3.6, the 5th step is by choice of $d$ (see Lemma 3.2), the 6th step is by $B = \Theta(k)$ (see Lemma 3.6).

**Proof of Running Time.**

By Lemma 3.6, the time complexity of HASHTOBINS is $O(k \log k)$. By Lemma 3.8, the time complexity of ONESPARSERECOVERY is $O(\log(F/\eta))$. Taking the median from $d$ results from HASHTOBINS takes time $O(d)$.

The time complexity of SUBLINEAR is

$d \log(F/\eta) \cdot \text{HASHTOBINS}$
$\quad + dB \cdot \text{ONESPARSERECOVERY}$
$\quad + \log(F/\eta) \cdot \text{Taking Median}$
$= d \log(F/\eta) \cdot k \log(k) + dB \cdot \log(F/\eta) + \log(F/\eta) \cdot d$
$= O(B \log(F/\eta)) \log(F/\eta) k \log(k)$
$\quad + O(B \log(F/\eta)) B \log(F/\eta)$
$\quad + \log(F/\eta) \cdot O(B \log(F/\eta))$
$= O(k^2 \log k \log^2(F/\eta))$

where the first step follows from the time complexity of HASHTOBINS, ONESPARSERECOVERY and Taking Median, the second step is by the choice of $d$, and the last step holds for $B = \Theta(k)$.

Hence, the time complexity for the main algorithm is

$$T^* \cdot O(k^2 \log k \log^2(F/\eta)) = O(k^2 \log k \log^3(F/\eta))$$

since $R^* = O((F/\eta)^m)$ by the assumption, which allows us to run only $T^* = O(\log(F/\eta))$ iterations of one-stage sparse recovery.    □

## 4. Conclusion

In this work, we study the deterministic algorithm for the sparse Fourier transform in the continuous setting, which bridges a significant gap in prior research dominated by randomized approaches. By leveraging innovative techniques such as deterministic hashing, robust noise modeling, and recursive sparse recovery algorithms, the proposed method achieves optimal recovery guarantees in sublinear time. This advancement not only extends the theoretical boundaries of sparse Fourier transforms but also paves the way for practical applications in signal processing, machine learning, and beyond, particularly in scenarios where deterministic solutions are desirable.

## Acknowledgements

The author would like to thank the anonymous reviewer of ICML 2025 for their highly insightful suggestions.

## Impact Statement

This paper presents work whose goal is to advance the field of Machine Learning and Signal Processing. There are many potential societal consequences of our work, but none of which we feel must be specifically highlighted here.

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

# Appendix

**Roadmap.** We organize Appendix as follow. In Section A, we introduce the de-randomization techniques which will be used to guarantee the success of sparse recovery algorithm. In Section B, we discuss the super-linear time sparse recovery algorithm. In Section C, we provide the missing proofs of our sub-linear time sparse recovery algorithm.

# A. De-randomization

In this section, our goal is to find a deterministic sequence of hashing $\{(\sigma_r, a_r, b_r)\}_{r \in [d]}$ satisfying the following condition (see Lemma A.21):

$$\sum_{r \in [d]} (\widehat{G}_{o_{f, \sigma_r, b_r}}(f))^{-1} \widehat{G}_{o_{f, \sigma_r, b_r}}(f') \leq \frac{\beta}{1 - \epsilon}.$$

This condition will be used to derive a guarantee for algorithm SUBRECOVERY (i.e., the fifth condition in Lemma B.2).

In Section A.1, we define several basic parameters and analyze the relationships between those parameters. In Section A.2, we provide the definition of sample range set. In Section A.3, we define the bad event. In Section A.4, we define Pessimistic Estimator. In Section A.5, we provide a round and mod tool. In Section A.6, we provide a lemma for offset function. In Section A.7, we provide an upper bound on $Z_\sigma(b)$. In Section A.8, we study the distribution of $\sigma(f' - f) \pmod 1$. In Section A.9, we study the distribution of $o$. In Section A.10, we provide bound for the range of $o$. In Section A.11, we prove the upper bound for probability of bad event. In Section A.12, we provide upper bound for $M(\lambda)$. In Section A.13, we analyze the initial constraint. In Section A.14, we analyze the induction steps. In Section A.15, we show how to put everything together.

## A.1. Parameter constraints

This section lists our choice for several parameters used in this section.

**Definition A.1.** *We define the following parameters:*

- $\lambda := C_1$ *where $C_1$ is some fixed constant belongs to $(\frac{1}{2}, 1)$*
- $\epsilon := \frac{20}{B}$
- $\beta := \frac{6}{C_1} \cdot \log |\mathcal{F}|$
- $d := \frac{3C_1}{40} \cdot B \log |\mathcal{F}|$

The parameters in Definition A.1 are chosen to satisfy below inequalities, which we will use in the proofs of this section.

**Observation A.2.** *We assume the parameters satisfy the following conditions:*

1. $\lambda \in (0, 1)$
2. $\epsilon \in (0, 1)$
3. $\beta \geq 4\epsilon d$
4. $\lambda \beta \geq 6 \log |\mathcal{F}|$
5. $\frac{10}{B} \cdot (\lambda(1 - \epsilon) + 1) \leq \lambda \epsilon$

*Proof.* **Proof of Part 5**

$$\frac{10}{B} \cdot (\lambda(1 - \epsilon) + 1) \leq \frac{10}{B} \cdot (\lambda + 1)$$
$$\leq \frac{\lambda \epsilon}{2} + \frac{10}{B}$$
$$\leq \lambda \epsilon$$

the first step is by $B > 20$, the 2nd step is by the choice of $\epsilon$, the 3rd step is by $\lambda > 1/2$ $\qquad\square$

### A.2. Sample Range Set

This section defines the set where we sample the hashing parameters $\sigma$ and $b$.

**Definition A.3** (Sample range of $b$). *Let $\mathcal{B}$ be the range of the hashing parameter $b$. We take $\mathcal{B} := \mathcal{F}$.*

**Definition A.4** (Sample range of $\sigma$). *Let $\Sigma$ be the range of the hashing parameter $\sigma$. We define $S$ to be the set of positive odd integers in $[0, F/\eta]$, and $\Sigma := \frac{1}{F} \cdot S = \{x/F \mid x \in S\}$.*

### A.3. Bad Event

This section defines the bad events for which two active frequencies are not filtered out.

**Definition A.5** (Bad events, Definition 5.4 in page 14 in Li & Nakos (2020)). *Suppose that*

- *$d$ is defined as Definition A.1, and*

- *$\beta$ is defined as Observation A.2.*

*For any $f, f' \in \mathcal{F}$, $f \neq f'$, we define $A_{f,f'}$ to be the event that*

$$\sum_{r=1}^{d} \widehat{G}_{o_{f,\sigma_r,b_r}(f')} \geq \beta.$$

### A.4. Pessimistic Estimator

This section defines the pessimistic estimator, which is used in choosing the property parameters in the hashing.

**Definition A.6** (Pessimistic estimator, Definition 5.5 in page 15 in Li & Nakos (2020)). *Let $\lambda > 0$ be a fixed parameter. We define the* pessimistic estimator *as follows:*

$$h_r(f, f'; \sigma_1, b_1, \cdots, \sigma_r, b_r) := \exp(-\lambda\beta) \cdot \exp\left(\lambda \sum_{l=1}^{r} \widehat{G}_{o_{f,\sigma_l,b_l}(f')}\right) \cdot (M(\lambda))^{d-r},$$

*where*

$$M(\lambda) := \exp(\lambda\epsilon) \cdot \left(\frac{5}{B} \cdot e^{\lambda(1-\epsilon)} + 1\right).$$

*For $r \geq 1$, the value of $\sigma_r, b_r$ is determined by the following minimization procedure:*

$$\sigma_r, b_r = \arg \min_{\sigma \in \Sigma, b \in \mathcal{B}} \sum_{f,f' \in \mathcal{F}: f \neq f'} h_r(f, f'; \sigma_1, b_1, \cdots, \sigma_{r-1}, b_{r-1}, \sigma, b).$$

*This function can be evaluated in $\mathrm{poly}(F/\eta)$ time.*

**Remark A.7.** *Li & Nakos (2020) chose $d$ to be $O(k \log n)$. In this work, we choose $d$ to be $O(k \log(F/\eta))$.*

### A.5. A Round and Mod Tool

This section introduces a technical tool to analyze the hashing function.

**Fact A.8** (Change order in taking modulus). *For any positive integer $y$ and real number $x$, it holds that*

$$(y^{-1} \cdot \mathrm{round}(yx)) \mod 1 = y^{-1} \cdot \mathrm{round}(y \cdot (x \mod 1)) - c,$$

*where $c = 0$ or $1$.*

*Proof.* We assume $x = x' + q$ where $x$ is an integer and $q \in [0, 1)$, then we have

$$(y^{-1} \cdot \mathrm{round}(yx)) \mod 1 = (y^{-1} \cdot \mathrm{round}(yx' + yq)) \mod 1$$

$$= (y^{-1} \cdot (yx' + \text{round}(yq))) \mod 1$$
$$= (y^{-1} \cdot \text{round}(yq)) \mod 1$$

where the first step is because $x = x' + q$, the 2nd step is because $yx'$ is an integer, the 3rd step is because $x' \mod 1 = 0$. Then, we consider two cases of $q$:

Case 1: $q \in [0, 1 - 1/(2y))$. Then, $yq < y - 1/2$, and $\text{round}(yq) \leq y - 1$. Thus, $y^{-1} \cdot \text{round}(yq) \in [0, 1)$. In this case,

$$\text{LHS} = (y^{-1} \cdot \text{round}(yq)) \mod 1$$
$$= (y^{-1} \cdot \text{round}(yq))$$
$$= y^{-1} \cdot \text{round}(y \cdot (x \mod 1)),$$

where the last step follows from $q = x \mod 1$. And the fact is proved with $c = 0$ in this case.

Case 2: $q \in [1 - 1/(2y), 1)$. Then, $yq \in [y - 1/2, y)$, and $\text{round}(yq) = y$. Then, $y^{-1} \cdot \text{round}(yq) = 1$. In this case,

$$\text{LHS} = (y^{-1} \cdot \text{round}(yq)) \mod 1 = 0.$$

On the other hand,

$$\text{RHS} = y^{-1} \cdot \text{round}(y \cdot (x \mod 1)) = (y^{-1} \cdot \text{round}(yq)) = 1.$$

Hence, the fact follows with $c = 1$ in this case.

As Case 1 and Case 2 consider all possible values of $q$, the fact is then proved. $\qquad\square$

### A.6. Reformulation of $o_{f,\sigma,b}(f')$

This section introduces a simplified version of the offset function $o_{f,\sigma,b}$.

**Lemma A.9** (Simplified $o_{f,\sigma,b}$). *Let $\pi_{\sigma,b}(f)$, $o_{f,\sigma,b}(f')$ be defined as Definition 2.4. We have*

$$o_{f,\sigma,b}(f') = \left( \sigma(f' - f) + \sigma(f - b) - \frac{1}{B} \text{round}(B\sigma(f - b)) \mod 1 \right) - c,$$

*for some $c = 0$ or $1$.*

*Proof.*

$$o_{f,\sigma,b}(f') = \pi_{\sigma,b}(f') - (1/B)h_{\sigma,b}(f) \mod 1$$
$$= \sigma(f' - b) \mod 1 - \frac{1}{B}\text{round}(B(\sigma(f - b) \mod 1))$$
$$= \sigma(f' - b) \mod 1 - \left( \left( \frac{1}{B}\text{round}(B\sigma(f - b)) \right) \mod 1 \right) - c$$
$$= \left( \left( \sigma(f' - b) - \frac{1}{B}\text{round}(B\sigma(f - b)) \right) \mod 1 \right) - c$$
$$= \left( \left( \sigma(f' - f) + \sigma(f - b) - \frac{1}{B}\text{round}(B\sigma(f - b)) \right) \mod 1 \right) - c,$$

where the first step and second step follows by Definition 2.4, the 3rd step uses Fact A.8 by taking $y = B$ and $x = \sigma(f - b)$, and $c = 0$ or $1$, the 4th step holds by property of taking modulus, the last step is a rearrangement.

$\qquad\square$

Notice that, the $-c$ will not affect the result showing below since $\sigma(f - f') \mod 1 \in [0, 1)$, so we ignore it.

**A.7. Upper bound on $Z_\sigma(b)$**

This section introduces an auxiliary variable $Z_\sigma(b)$, which is used to control the distribution of the offset hashing function.

**Definition A.10.** *Suppose $b$ is uniformly sampled from the set $\mathcal{B}$ (Definition A.3), and $\sigma$ is uniformly sampled from the set $\Sigma$ (Definition A.4). Define*

$$Z_\sigma(b) := \sigma(f - b) - \frac{1}{B}\mathrm{round}(B\sigma(f - b)).$$

The next lemma upper bounds $Z_\sigma(b)$.

**Lemma A.11** (The range of $Z_\sigma(b)$). *Let $Z_\sigma(b)$ be defined as in Definition A.10. Then, it always holds that*

$$Z_\sigma(b) \in [-\frac{1}{2B}, \frac{1}{2B}].$$

*Proof.* We can show

$$
\begin{aligned}
|Z_\sigma(b)| &= |\sigma(f - b) - \frac{1}{B}\mathrm{round}(B\sigma(f - b))| \\
&= \frac{1}{B} \cdot |B\sigma(f - b) - \mathrm{round}(B\sigma(f - b))| \\
&\leq \frac{1}{2B}
\end{aligned}
$$

where the first step is by definition of $Z_\sigma(b)$ (see Definition A.10), the 2nd step is from simple calculation, the 3rd step holds because $|x - \mathrm{round}(x)| \leq 1/2$ for all $x \in \mathbb{R}$.

$\square$

**A.8. Distribution of $\sigma(f' - f) \pmod 1$**

This section analyzes the distribution of $\sigma(f' - f) \pmod 1$. We begin with a simplified version of $\sigma(f' - f)$.

**Lemma A.12** (Reformulation of $\sigma(f' - f)$). *Under the following conditions:*

- *Let $\sigma$ be uniformly random from set $\Sigma$ (see Definition A.4)*

- *Let $m := \sigma F$*

- *Let $i := (f' - f)/\eta$*

*then $m$ is uniformly distributed over $S$ (see Definition A.4), and*

$$(\sigma(f' - f)) \mod 1 = \frac{\eta}{F} \cdot (mi \mod \frac{F}{\eta}). \tag{1}$$

*Proof.* By Definition A.4), we know that $\Sigma = \frac{1}{F} \cdot S$, where $S$ contains all the odd numbers on $[F/\eta]$. Since $\sigma$ is uniformly sampled from $\Sigma$ and $m = \sigma F$, we get that $m$ is uniformly distributed in $S$.

By the definitions of $m$ and $i$, we have

$$\sigma(f' - f) = \frac{m}{F} \cdot i\eta = mi \cdot \frac{\eta}{F}.$$

Now suppose $\sigma(f' - f) = C + D$, where $C \in \mathbb{Z}$, $D \in [0, 1)$. Then, we have

$$mi \cdot \frac{\eta}{F} = C + D, \quad \text{i.e.,} \quad mi = C\frac{F}{\eta} + D\frac{F}{\eta}.$$

Since $D\frac{F}{\eta} \in [0, \frac{F}{\eta})$, it implies that

$$mi \quad \mathrm{mod} \ \frac{F}{\eta} = D\frac{F}{\eta}.$$

Now, we can conclude that:

$$(\sigma(f' - f)) \quad \mathrm{mod} \ 1 = D = \frac{\eta}{F} \cdot (mi \quad \mathrm{mod} \ \frac{F}{\eta}).$$

□

The next statement characterizes the distribution of $(\sigma(f' - f) \mod 1)$.

**Lemma A.13** (Distribution of $(\sigma(f' - f) \mod 1)$). *Under the following conditions,*

- *Suppose $F/\eta := 2^p$, where $p$ is a positive integer*

- *Let $\sigma$ be uniformly random from set $\Sigma$ (see Definition A.4)*

- *Let $f' \neq f$ be frequencies from $\mathcal{F}$*

- *Let $m, i$ be defined as in Lemma A.12*

- *Let $i := 2^s K$ where $s$ is a non-negative integer and $K$ is an odd number*

*then we have*

1. *$(\sigma(f' - f) \mod 1)$ is uniformly distributed over its support*

2. *The support of $(\sigma(f' - f) \mod 1)$ is symmetric*

3. *The support of $(\sigma(f' - f) \mod 1)$ is a sequence of equidistant points, with wraparound distance $D := \frac{\eta}{F}2^{s+1}$*

*In particular, $(\sigma(f' - f) \mod 1)$ is uniformly distributed over the following set:*

$$\left\{ \pm(\frac{1}{2} + j) \cdot D : 0 \leq j \leq 2^{p-s-2} - 1, j \in \mathbb{N} \right\}.$$

*Proof.* By Lemma A.12, $(\sigma(f' - f) \mod 1) = 2^{-p} \cdot (mi \mod 2^p)$. Hence, we only need to consider the distribution of

$$mi \quad (\mathrm{mod} \ 2^p),$$

where $m$ is uniformly sampled from odd integers in $[2^p]$ and $i = 2^s K \in [2^p]$.

If $i = 2^p$, then $\mathrm{supp}(\sigma(f' - f) \mod 1) = \{0\}$. And the lemma trivially holds. In the following proof, we assume that $i < 2^p$, i.e., $0 \leq s < p$.

**Proof of Part 1**

Let $m_1, m_2$ be two possible value of $m$, when $m_1 - m_2 = 2^{p-s}$, we have

$$m_1 i - m_2 i = 2^p K \equiv 0 \quad (\mathrm{mod} \ 2^p)$$

Hence, the value of $(mi \mod 2^p)$ has a period of length at least $2^{p-s}$. And we only need to consider $m \in [2^{p-s}]$.

Let $m_1, m_2 \in [2^{p-s}], m_1 > m_2$. Then

$$(m_1 - m_2)2^s < 2^p$$

Since $K$ is odd, then $(m_1 - m_2)i = (m_1 - m_2)2^s K$ cannot be divided by $2^p$, which imples that

$$m_1 i \pmod{2^p} \neq m_2 i \pmod{2^p}$$

Therefore, each odd integer $m \in [2^{p-s}]$ generates a unique value for $mi \pmod{2^p}$. And $mi = m'i \pmod{2^p}$ for every $m' \in \{m + j2^{p-s} \mid m + j2^{p-s} \leq 2^p, j \in \mathbb{N}\}$. Suppose that $i < 2^p$, then each $m'$ in this set is an odd number. Moreover, for any odd $m \in [2^{p-s}]$,

$$|\{m + j2^{p-s} \mid m + j2^{p-s} \leq 2^p, j \in \mathbb{N}\}| = 2^s.$$

Therefore, $S$, the support of $m$, is divided into $2^s$-sized equivalence classes, and each class gives a distinct value for $mi$ $(\mod 2^p)$. Since $m$ is uniformly sampled, $mi \pmod{2^p}$ is uniform on its support, so does $\sigma(f' - f) \pmod 1$.

**Proof of Part 2**

For any $m \in [2^{p-s}]$, we have

$$mi + (2^{p-s} - m)i = 2^p K \equiv 0 \pmod{2^p}.$$

Therefore,

$$(mi \mod 2^p) + ((2^{p-s} - m)i \mod 2^p) = 2^p,$$

that is, the support of $(mi \mod 2^p)$ is symmetric in $[2^p]$. Thus, the support of $(\sigma(f' - f) \mod 1)$ is symmetric in $[0, 1]$ with respect to $0$ under wraparound distance.

**Proof of Part 3**

For any $m_1 > m_2 \in [2^{p-s}]$, $m_1 - m_2$ can be written as $2g$ for some integer $g \neq 0$ since they are all odd numbers. We have

$$(m_1 - m_2)i = 2^{s+1} gK = C \cdot 2^p + D$$

where $C$ is an integer and $D \in (0, 2^p)$. Then, we know that $D$ can be divided by $2^{s+1}$ since $2^{s+1}gK$ and $C \cdot 2^p$ both can be divided by $2^{s+1}$.

We show that $D$ is the wraparound distance between any two points in the support of $(mi \mod 2^p)$:

$$\begin{aligned}
&((m_1 i \pmod{2^p})) - (m_2 i \pmod{2^p}) \mod 2^p) \\
&= (m_1 - m_2)i \mod 2^p \\
&= 2^{s+1}gK \mod 2^p \\
&= D,
\end{aligned}$$

where the first step is by the property of taking modulus, the 2nd step is because $m_1 - m_2 = 2g$, and the 3rd step follows from the definition of $D$. Then, we know that the wraparound distance between two points is at least $2^{s+1}$.

On the other hand, we show that the distance is at most $2^{s+1}$. By **Part 1** of this lemma, we know that

$$|\{mi \pmod{2^p} : m \in [2^p], m \text{ is odd}\}| = |\{m \in [2^{p-s}], m \text{ is odd}\}| = 2^{p-s-1}.$$

Thus,

$$D \cdot 2^{p-s-1} \leq 2^p,$$

which implies that $D \leq 2^{s+1}$.

Therefore, we get that the support of $(mi \mod 2^p)$ is a sequence of equidistant points, with wraparound distance $2^{s+1}$. By scaling a factor of $\frac{\eta}{F}$, we get the wanted result. $\qquad \square$

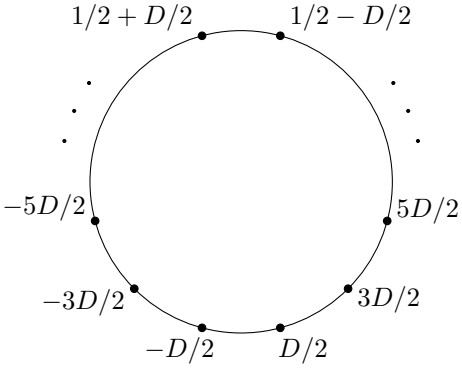

*Figure 1.* The support of $(\sigma(f' - f) \mod 1)$, where $D = \frac{\eta}{F}2^{s+1}$.

## A.9. Distribution of $o_{f,\sigma,b}(f')$

This section analyzes the distribution of $o_{f,\sigma,b}(f')$. The next statement shows that the offset of $f'$ with respect to $f$ is large with high probability.

**Lemma A.14** (Analogous to Lemma 5.6 in page 15 in Li & Nakos (2020)). *If the following conditions hold:*

- *Suppose $F/\eta := 2^p$, where $p$ is a positive integer*

- *Let $\sigma$ be uniformly random from set $\Sigma$ (see Definition A.4)*

- *Let $f' \neq f$ be frequencies from $\mathcal{F}$*

- *Let $m, i$ be defined as in Lemma A.12*

- *Let $i := 2^s K$ where $s$ is a non-negative integer and $K$ is an odd number*

- *Suppose $B = 2^q$*

*then for any $f \neq f'$ we have,*

$$\Pr_{\sigma,b}\left[o_{f,\sigma,b}(f') \in \left[-\frac{1}{B}, \frac{1}{B}\right]\right] \leq \frac{5}{B}.$$

*Proof.* By Lemma A.9, we know that

$$o_{f,\sigma,b}(f') = \sigma(f' - f) + Z_\sigma(b) \mod 1,$$

where $Z_\sigma(b)$ is a random variable defined by

$$Z_\sigma(b) = \sigma(f - b) - \frac{1}{B}\text{round}(B\sigma(f - b)).$$

Then, the distribution of $o_{f,\sigma,b}(f')$ is a convolution of $Z_\sigma(b) \in [-\frac{1}{2B}, \frac{1}{2B}]$ and $\sigma(f' - f) \mod 1$ (a sequence of equidistant points).

We first consider the random variable $\sigma$, and then conditioned on $\sigma$, we consider the random variable $b$.

By Lemma A.13, the distance between two consecutive points in the support of $\sigma(f' - f) \mod 1$ is $D = \frac{\eta}{F}2^{s+1} = 2^{-p+s+1}$. In the following proof, we discuss two cases based on the value of $p - q - s$.

**Case 1:** $p - q \geq s$. We have

$$\frac{1}{B} = 2^{-q} = D \cdot 2^{p-q-s-1} \geq \frac{D}{2}. \tag{2}$$

Then, we have

$$\Pr\left[o_{f,\sigma,b}(f') \in \left[-\frac{1}{B}, \frac{1}{B}\right]\right]$$

$$= \Pr\left[(\sigma(f' - f) + Z_\sigma(b) \mod 1) \in \left[-\frac{1}{B}, \frac{1}{B}\right]\right]$$

$$= \sum_{x \in \text{supp}(\sigma(f'-f) \mod 1)} \Pr_b\left[(x + Z_\sigma(b) \mod 1) \in \left[-\frac{1}{B}, \frac{1}{B}\right] \,\Big|\, (\sigma(f' - f) \mod 1) = x\right] \cdot$$

$$\Pr_\sigma[(\sigma(f' - f) \mod 1) = x]$$

$$\leq \Pr\left[(\sigma(f' - f) \mod 1) \in \left[-\frac{3}{2B}, \frac{3}{2B}\right]\right]$$

$$\leq \frac{(1/D) \cdot (3/B) + 1}{1/D}$$

$$= \frac{3}{B} + D$$

$$\leq \frac{5}{B}$$

where the 1st step is by simple algebra, the 2nd step follows from the conditional probability, the 3rd step is because $Z_\sigma(b) \in [-\frac{1}{2B}, \frac{1}{2B}]$ (see Lemma A.11), the 4th step is because the uniform distribution of $\sigma(f' - f) \mod 1$ on its support described in Lemma A.13, where the numerator is the maximum number of points inside interval $[-\frac{3}{2B}, \frac{3}{2B}]$, the denominator is the total number of points in the whole range, the 5th step is a rearrangement, and the last step uses Eq. (2).

**Case 2:** $p - q \leq s - 1$. By Lemma A.13, the closest point to the origin in the support of $\sigma(f' - f) \mod 1$ is $\pm D/2$. Then, in this case, it holds that

$$\frac{D}{2} = 2^{-p+s} = \frac{2}{B} \cdot 2^{q-1-p+s} \geq \frac{2}{B} > \frac{3}{2B}.$$

Hence, by the same analysis as in Case 1, $o_{f,\sigma,b}(f')$ will not take value in $[-\frac{1}{B}, \frac{1}{B}]$.

$\square$

The next statement upper bounds the moment generation function of $\widehat{G}_{o_{f,\sigma,b}(f')}$, which is used to bound the probability of bad event.

**Lemma A.15** (Analogous to Lemma 5.7 in page 16 in Li & Nakos (2020)). *Under following conditions*

- *Suppose $|\mathcal{F}|$ is power of 2*

- *Let $\sigma$ be uniformly random from set $\Sigma$ (see Definition A.4)*

- *Let $b$ be uniformly random in set $\mathcal{B}$ (see Definition A.3)*

- *Suppose $B$ is power of 2*

- *Let $\widehat{G}$ be a flat filter defined in Definition 2.8*

- *Let $M(\lambda)$ be defined as Definition A.6*

*For all $f, f' \in \mathcal{F}$,*

$$\mathbb{E}_{\sigma,b}[\exp(\lambda \widehat{G}_{o_{f,\sigma,b}(f')})] \leq M(\lambda)$$

*Proof.* By Lemma A.14 we have,

$$\Pr[\widehat{G} \geq \epsilon] = \Pr[o_{f,\sigma,b}(f') \in [-\frac{1}{B}, \frac{1}{B}]] \leq \frac{5}{B} \tag{3}$$

where the first step is by the definition of $G$ (see Definition 2.8), the second step uses **Part 1** of Lemma A.14.

Therefore, we have

$$\mathbb{E}_{\sigma,b}[\exp(\lambda \widehat{G}_{o_{f,\sigma,b}(f')})] \leq \Pr[\widehat{G} \geq \epsilon] \cdot \exp(\lambda \cdot \sup \widehat{G}) + \Pr[\widehat{G} \leq \epsilon] \cdot \exp(\lambda \epsilon)$$

$$\leq \Pr[\widehat{G} \geq \epsilon] \cdot \exp(\lambda) + \Pr[\widehat{G} \leq \epsilon] \cdot \exp(\lambda \epsilon)$$

$$\leq \frac{5}{B} \cdot e^{\lambda} + e^{\lambda \epsilon}$$

$$= M(\lambda)$$

where the first step follows from the definition of expectation, the second step follows by $\widehat{G} \in [0, 1]$ (see Definition 2.8), the 3rd step is by Eq. (3), and the last step follows from the definition of $M(\lambda)$ (see Definition A.6).

$\square$

## A.10. Range of $o_{f,\sigma,b}(f)$

This section bounds the value of $o_{f,\sigma,b}(f)$.

**Lemma A.16.** *If the following conditions hold:*

- *Let $\sigma$ be randomly chose from set $\Sigma$ (see Definition A.4)*

- *Let $b$ be uniformly random from set $\beta$ (see Definition B.6)*

*then for any $f \in \mathcal{F}$ we have,*

$$o_{f,\sigma,b}(f) \in [-\frac{1}{2B}, \frac{1}{2B}]$$

*Proof.*

$$o_{f,\sigma,b}(f) = \sigma(f - f) + Z_{\sigma}(b)$$

$$= Z_{\sigma}(b)$$

$$\in [-\frac{1}{2B}, \frac{1}{2B}]$$

where the first step is from Lemma A.9, the 3rd step is by Lemma A.11

$\square$

## A.11. Upper bound for probability of bad event

This section upper bounds the probability of bad event by the pessimistic estimator.

**Lemma A.17** (Pessimistic Estimator, Analogous to Lemma 5.8 in page 16 in Li & Nakos (2020)). *Under following conditions*

- *Suppose $|\mathcal{F}|$ is power of 2*

- *Let $r \in [d]$*

- *Let $\sigma$ be uniformly random from set $\Sigma$ (see Definition A.4)*

- *Let $b$ be uniformly random in set $\mathcal{B}$ (see Definition A.3)*

- *Let $h_r$ be defined as Definition A.6*

- *Let $H_r = \{(\sigma_r, a_r, b_r)\}_{r \in [d]}$ be a sequence of hashing chose by procedure in Definition A.6*

- *Suppose $f, f' \in \mathcal{F}$ satisfy $f \neq f'$*

- *Let $A_{f,f'}$ denote the bad event defined as Definition A.5, where $(\sigma_{r+1}, b_{r+1}), \ldots, (\sigma_d, b_d)$ are uniformly and independently sampled from $\Sigma \times \mathcal{B}$*

*we have,*

$$h_r(f, f'; \sigma_1, b_1, \cdots, \sigma_r, b_r) \geq \Pr[A_{f,f'} \mid \sigma_1, b_1, \ldots, \sigma_r, b_r].$$

*Proof.* We define $z$ as follows,

$$z := \sum_{l=1}^{r} \widehat{G}_{o_{f,\sigma_l,b_l}}(f')$$

Conditioned on $\sigma_1, b_1, \ldots, \sigma_r, b_r$, $z$ is a fixed constant.

Then we have

$$
\begin{aligned}
\Pr[A_{f,f'} \mid \sigma_1, b_1, \cdots, \sigma_r, b_r] &= \Pr[z + \sum_{l=r+1}^{d} \widehat{G}_{o_{f,\sigma_l,b_l}}(f') > \beta] \\
&= \Pr[\exp(\lambda(z + \sum_{l=r+1}^{d} \widehat{G}_{o_{f,\sigma_l,b_l}}(f'))) > e^{\lambda \beta}] \\
&\leq e^{-\lambda \beta} e^{\lambda z} \, \mathbb{E}[\exp(\lambda \sum_{l=r+1}^{d} \widehat{G}_{o_{f,\sigma_l,b_l}}(f'))] \\
&= e^{-\lambda \beta} e^{\lambda z} \, \mathbb{E}[\exp(\lambda \widehat{G}_{o_{f,\sigma,b}}(f'))]^{d-r} \\
&= e^{-\lambda \beta} e^{\lambda z} (M(\lambda))^{d-r}
\end{aligned}
$$

where the 3rd step is by Markov inequality, the 4th step follows from the independence, the 5th step is given by Lemma A.15, and the expression in the last line is exactly $h_r$ (see Definition A.6). $\qquad\square$

### A.12. Upper bound for $M(\lambda)$

This section upper bounds the quantity $M(\lambda)$.

**Lemma A.18** (Upper bound for $M(\lambda)$)**.** *Under following conditions*

- *Let $M(\lambda)$ be defined as in Definition A.6*

- *Suppose that $\epsilon \in (0, 1)$*

*Then we have,*

$$M(\lambda) \leq \exp(2\lambda\epsilon)$$

*Proof.*

$$
\begin{aligned}
M(\lambda) &= \exp(\lambda\epsilon) \cdot (\frac{5}{B} \cdot e^{\lambda(1-\epsilon)} + 1) \\
&\leq \exp\left(\lambda\epsilon + \log(1 + \frac{5}{B} \cdot e^{\lambda(1-\epsilon)})\right)
\end{aligned}
$$

$$\leq \exp\left(\lambda\epsilon + \frac{5}{B} \cdot e^{\lambda(1-\epsilon)}\right)$$

$$\leq \exp\left(\lambda\epsilon + \frac{10}{B} \cdot (\lambda(1-\epsilon)+1)\right)$$

$$\leq \exp(2\lambda\epsilon)$$

where the 1st step is due to definition of $M(\lambda)$ (see Definition A.6), the 2nd is by simple algebra, the 3rd step is because $\log(x+1) \leq x$ for $x \neq 0$, the 4th step is because $e^x \leq 2x + 1$ for $x \in (0,1)$, and $\lambda, \epsilon \in (0,1)$, the 5th step is by **Part 5** of Observation A.2

$\square$

## A.13. Initial Constraint

The following sections upper bound the pessimistic estimator by induction. This section verifies the initial constraint.

**Lemma A.19** (Initial constraint, Analogous to Lemma 5.9 in page 17 in Li & Nakos (2020)). *Under following conditions*

- *Suppose $|\mathcal{F}|$ is power of 2*

- *Let $\sigma$ be uniformly random from set $\Sigma$ (see Definition A.4)*

- *Let $b$ be uniformly random in set $\mathcal{B}$ (see Definition A.3)*

- *Let $h_r$ be defined as Definition A.6*

- *Suppose $f, f \in \mathcal{F}$ satisfy $f \neq f'$*

*we have,*

$$\sum_{f,f' \in \mathcal{F}: f \neq f'} h_0(f, f') < 1$$

*Proof.*

$$\sum_{f,f' \in \mathcal{F}: f \neq f'} h_0(f, f') = e^{-\lambda\beta} \sum_{f,f' \in \mathcal{F}: f \neq f'} (M(\lambda))^d$$

$$\leq |\mathcal{F}|^2 \exp(-\lambda\beta + 2\lambda\epsilon d)$$

$$\leq |\mathcal{F}|^2 \exp(-0.5\lambda\beta)$$

$$\leq |\mathcal{F}|^2 \exp(-3\log|\mathcal{F}|)$$

$$< 1$$

where the first step is from definition of $h_0$ (see Definition A.6), the second step follows from Lemma A.18, the third step is by $\beta \geq 4\epsilon d$ (**Part 3** of Observation A.2), the fourth step follows by $\lambda\beta \geq 6\log|\mathcal{F}|$) (**Part 4** of Observation A.2), the fifth step is by simple algebra. $\square$

## A.14. Induction step

This section shows the induction step.

**Lemma A.20** (Derandomization, Analogous to Lemma 5.10 in page 17 in Li & Nakos (2020)). *Under following conditions*

- *Suppose $|\mathcal{F}|$ is power of 2*

- *Let $\sigma$ be uniformly random from set $\Sigma$ (see Definition A.4)*

- *Let $b$ be uniformly random in set $\mathcal{B}$ (see Definition A.3)*

- *Let $r \in [d-1]$*

- *Let $H_r = \{(\sigma_j, a_j, b_j)\}_{j \in [j]}$ be a sequence of hashing chose by procedure in Definition A.6*

- *Let $h_r$ be defined as Definition A.6*

- *Suppose $f, f \in [n]$ satisfy $f \neq f'$*

*we have,*

$$h_r(f, f'; \sigma_1, b_1, \cdots, \sigma_r, b_r) \geq \underset{\sigma_{r+1}, b_{r+1}}{\mathbb{E}}[h_{r+1}(f, f'; \sigma_1, b_1, \cdots, \sigma_{r+1}, b_{r+1})]$$

*Proof.* We define $z$ as follows,

$$z := \sum_{l=1}^{r} \widehat{G}_{o_{f,\sigma_l, b_l}}(f')$$

Then we have

$$
\begin{aligned}
\underset{\sigma_{r+1}, b_{r+1}}{\mathbb{E}}[h_{r+1}(f, f'; \sigma_1, b_1, \cdots, \sigma_{r+1}, b_{r+1})] &= \underset{\sigma, b}{\mathbb{E}}[e^{-\lambda\beta}e^{\lambda(z + \widehat{G}_{o_{f,\sigma,b}}(f'))}(M(\lambda))^{d-r-1}] \\
&= e^{-\lambda\beta}e^{\lambda z}(M(\lambda))^{d-r-1}\underset{\sigma, b}{\mathbb{E}}[\exp(\lambda\widehat{G}_{o_{f,\sigma,b}}(f'))] \\
&\leq e^{-\lambda\beta}e^{\lambda z}(M(\lambda))^{d-r} \\
&= h_r(f, f'; \sigma_1, b_1, \cdots, \sigma_r, b_r)
\end{aligned}
$$

where the 1st step is due to definition of $h_r$ (see Definition A.6), the 2nd step holds since $e^{\lambda z}$ and $M(\lambda)$ are independent of $\sigma, b$, the 3rd step uses Lemma A.15, the 4th step is due to definition of $h_r$ (see Definition A.6). $\qquad\square$

### A.15. Putting it all together

This section summarizes the analysis above and shows our final result.

**Lemma A.21.** *If the following happens*

- *Let $\beta$ be defined as Definition A.1*

- *Let $\epsilon \in (0, 1)$ be defined as Definition A.1*

- *Let $H_d = \{(\sigma_r, a_r, b_r)\}_{r \in [d]}$ be a sequence of hashing chose by procedure in Definition A.6*

- *Let $\widehat{G}$ be a flat filter in accordance of hashing functions in $H_r$ (see Definition 2.8)*

*it holds that, for all $f \neq f'$*

$$\sum_{r \in [d]} \widehat{G}_{o_{f,\sigma_r, b_r}}^{-1}(f)\widehat{G}_{o_{f,\sigma_r, b_r}}(f') \leq \frac{\beta}{1 - \epsilon}$$

*Proof.* Note that in each step, we choose $\sigma_{r+1}, b_{r+1}$ to minimize

$$\sum_{f,f' \in \mathcal{F}: f \neq f'} h_{r+1}(f, f'; \sigma_1, b_1, \cdots, \sigma_{r+1}, b_{r+1}).$$

Then, we know that

$$\sum_{f,f' \in \mathcal{F}: f \neq f'} h_{r+1}(f, f'; \sigma_1, b_1, \cdots, \sigma_{r+1}, b_{r+1}) \leq \sum_{f,f' \in \mathcal{F}: f \neq f'} \underset{\sigma'_{r+1}, b'_{r+1}}{\mathbb{E}}[h_{r+1}(f, f'; \sigma_1, b_1, \cdots, \sigma'_{r+1}, b'_{r+1})], \quad (4)$$

which follows from the linearity of expectation. By Lemma A.20, it holds that

$$\sum_{f,f' \in \mathcal{F}: f \neq f'} \mathop{\mathbb{E}}_{\sigma'_{r+1}, b'_{r+1}} [h_{r+1}(f, f'; \sigma_1, b_1, \cdots, \sigma'_{r+1}, b'_{r+1})] \leq \sum_{f,f' \in \mathcal{F}: f \neq f'} h_r(f, f'; \sigma_1, b_1, \cdots, \sigma_r, b_r).$$

Hence by induction, we have

$$\sum_{f,f' \in \mathcal{F}: f \neq f'} h_d(f, f'; \sigma_1, b_1, \cdots, \sigma_d, b_d) \leq \sum_{f,f' \in \mathcal{F}: f \neq f'} h_0(f, f') < 1$$

where the 2nd step is given by Lemma A.19.

Therefore, by Lemma A.17,

$$\sum_{f,f' \in \mathcal{F}: f \neq f'} \Pr[A_{f,f'} \mid \sigma_1, b_1, \cdots, \sigma_d, b_d] \leq \sum_{f,f' \in \mathcal{F}: f \neq f'} h_d(f, f'; \sigma_1, b_1, \cdots, \sigma_d, b_d) < 1.$$

Note that conditioned on $\sigma_1, b_1, \cdots, \sigma_d, b_d$, $A_{f,f'}$ is a deterministic event. That is, the conditional probability for each pair of $f \neq f'$ is either zero or one. By the inequality, we get that

$$\Pr[A_{f,f'} \mid \sigma_1, b_1, \cdots, \sigma_d, b_d] = 0 \quad \forall f \neq f' \in \mathcal{F}.$$

Then by the definition of $A_{f,f'}$ (Definition A.5), it implies that

$$\sum_{r \in [d]} \widehat{G}_{o_{f,\sigma_r,b_r}}(f') \leq \beta. \tag{5}$$

Note that $o_{f,\sigma_r,b_r}(f) \in [-\frac{1}{2B}, \frac{1}{2B}]$ (see Lemma A.16). Thus, by definition of $\widehat{G}$ (see Definition 2.8), we have

$$\widehat{G}_{o_{f,\sigma_r,b_r}}(f) \in [1 - \epsilon, 1].$$

Then Eq. (5) gives that, for all $f \neq f' \in \mathcal{F}$,

$$\sum_{r \in [d]} \widehat{G}^{-1}_{o_{f,\sigma_r,b_r}}(f) \widehat{G}_{o_{f,\sigma_r,b_r}}(f') \leq \frac{\beta}{1 - \epsilon}.$$

The lemma is then proved. $\qquad\square$

## B. Super-linear Algorithm

This section gives a deterministic super-linear algorithm to recover the $k$ heavy-hitters. We first analyze the guarantee of HASHTOBINS, and then we embed it into a recursive algorithm that filters the signal in each bin by a decaying threshold.

In Section B.1, we show that the measurement gives a close approximation to the tones. In Section B.2, we present the assumption of the noise function. In Section B.3, we introduce HashToBins. In Section B.4, we analyze the median of the output of HashToBins. In Section B.5, we study the guarantee of the algorithm for super-linear time sparse recovery, which is an essential subroutine. In Section B.6, we analyze the guarantee and running time of the super-linear time main algorithm. In Section B.7, we analyze our result for the deterministic continuous Fourier transform.

### B.1. The Guarantees of Measurement

According to our choice of parameters in Definition A.1, we have the below relationship between $d/\beta$ and $B$. This allows us to normalize the $\ell_1$ bound of our algorithm by a factor of $1/k$.

**Observation B.1.** *Let* $\beta, d, \epsilon$ *be chosen as Definition A.1, we have*

$$\frac{\beta}{(1 - \epsilon)d} = \Theta(\frac{1}{B})$$

*Proof.*

$$\frac{\beta}{(1-\epsilon)d} = \frac{\Theta(\log|\mathcal{F}|)}{(1-\Theta(1/B)) \cdot \Theta(B\log(|\mathcal{F}|))} = \Theta(\frac{1}{B})$$

where the above equation is due to Definition A.1. □

The next statement shows the guarantee of the measurement.

**Lemma B.2** (A variation of Lemma 5.1 in page 11 in Li & Nakos (2020)). *Under following conditions*

- *Let $H_r = \{\sigma_r, a_r, b_r\}_{r \in d}$ be a sequence of hashing defined in Definition 2.6*

- *We have $\widehat{G}$ being a flat filter with $B$ buckets and sharpness $\epsilon$ (see Definition 2.8)*

- *Let $\widehat{x}, v$ be defined as Definition 1.1*

- *For all $f \neq f' \in \mathcal{F}$ it holds that,*

$$\sum_{r \in [d]} \widehat{G}^{-1}_{o_{f,\sigma_r,b_r}(f)} \widehat{G}_{o_{f,\sigma_r,b_r}}(f') \leq \frac{\beta}{1-\epsilon}$$

  *where $\beta$ is chosen as Definition A.1.*

*Then for every vector $x : [0, T] \to \mathbb{C}^n$ and every $f \in \mathcal{F}$, for at least $0.8d$ indices $r \in [d]$ we have*

$$|v_f - \widehat{G}^{-1}_{o_{f,\sigma_r,b_r}(f)}(m_{H_r})_{h_{\sigma_r,b_r}(f)}\omega^{-a_r\sigma f}| \leq \Theta(\frac{1}{B}) \cdot \sum_{f' \in \mathcal{F}\setminus\{f\}} |v_{f'}| \tag{6}$$

*Proof.* The proof is close to Li & Nakos (2020), we keep it here for completeness.

$$\sum_{r \in [d]} |v_f - \widehat{G}^{-1}_{o_{f,\sigma_r,b_r}(f)}(m_{H_r})_{h_r(f)}\omega^{-a\sigma f}|$$

$$= \sum_{r \in [d]} |\widehat{G}^{-1}_{o_{f,\sigma_r,b_r}(f)} \sum_{f' \in \mathcal{F}\setminus\{f\}} \widehat{G}_{o_{f,\sigma_r,b_r}}(f') v_{f'}\omega^{a_r\sigma_r(f'-f)}|$$

$$\leq \sum_{r \in [d]} \widehat{G}^{-1}_{o_{f,\sigma_r,b_r}(f)} \sum_{f' \in \mathcal{F}\setminus\{f\}} \widehat{G}_{o_{f,\sigma_r,b_r}}(f')|v_{f'}|$$

$$= \sum_{f' \in \mathcal{F}\setminus\{f\}} |v_{f'}| \sum_{r \in [d]} \widehat{G}^{-1}_{o_{f,\sigma_r,b_r}(f)}\widehat{G}_{o_{f,\sigma_r,b_r}}(f')$$

$$\leq \sum_{f' \in \mathcal{F}\setminus\{f\}} |v_{f'}|\frac{\beta}{1-\epsilon}$$

where the first step uses Claim 2.11, the 2nd step is given by triangle inequality, the 3rd step is a change of summation order, the 4th step is by the condition of this lemma. the last step is by the Observation B.1.

Therefore, at most $\frac{1}{5}$ fraction of $r \in [d]$ satisfy

$$|v_f - \widehat{G}^{-1}_{o_{f,\sigma_r,b_r}(f)}(m_{H_r})_{h_{\sigma_r,b_r}(f)}\omega^{-a\sigma f}| > \frac{5\beta}{(1-\epsilon)d} \cdot \sum_{f' \in \mathcal{F}\setminus\{f\}} |v_{f'}|$$

$$= \Theta(\frac{1}{B}) \cdot \sum_{f' \in \mathcal{F}\setminus\{f\}} |v_{f'}|,$$

where the last step is by Observation B.1.

Thus, we complete the proof. □

### B.2. Assumption of noise function

This section introduces the definition of $(C, \xi)$-noise function. We will show its restriction on the output of HASHTOBINS algorithm in the next section.

**Definition B.3** $((C, \xi)$-noise$)$. *Under following conditions*

- *Let $g(t) : [0, T] \to \mathbb{R}$ be the noise function*

- *Let $C$ be a fixed constant*

- *Let $\xi$ be a parameter depend on $g(t)$*

*then we say $g(t)$ is a $(C, \xi)$-noise if it satisfies*

$$\max_{t \in [0,T]} |g(t)|^2 \leq C \cdot \frac{1}{T} \int_0^T |g(t)|^2 \mathrm{d}t + \xi$$

### B.3. Hash to bins

This section presents the deterministic HASHTOBINS in the continuous setting. The algorithm is displayed in Algorithm 1. The next lemma states the identities of the pseudo-random permutation and the output of HASHTOBINS.

**Lemma B.4** (Identities of DFT and CFT, Lemma 4.3 and Fact 4.1 in Jin et al. (2023)). *Under following conditions*

- *Let $P_{\sigma,a,b}x$ be defined as Definition 2.5*

- *Let $x^*(t)$ be the noiseless signal*

- *Let $z(t) := \sum_{f \in \mathcal{F}} \widehat{z}_f \cdot e^{2\pi \mathbf{i} f t}$*

- *Let $\widehat{u}$ be the output of HASHTOBINS*

*we have*

- *Property 1: Identity of pseudo-permutation:*

$$\widehat{P_{\sigma,a,b}x^*}(t) = \frac{1}{\sigma}\widehat{x}^*(\frac{t}{\sigma} + b)e^{2\pi \mathbf{i} a(t + b\sigma)}$$

- *Property 2: Identity of output of HASHTOBINS: For any $j \in [B]$*

$$\widehat{u}_j = \widehat{G} * \widehat{P_{\sigma,a,b}(x^* - z)}(j/B)$$

*Proof.* Notice that **Property 2** in Jin et al. (2023) contains only $x$. However, we can extend it to our result by linear operation. $\qquad \square$

The next lemma shows the guarantee of the HASHTOBINS procedure under the noiseless version and the noise-only version.

**Lemma B.5** (HASHTOBINS). *If the following conditions hold:*

- *let $H = (\sigma, a, b)$ be a tuple of hashing defined in Definition 2.4*

- *Let $\widehat{G}$ be a flat filter (see Definition 2.8)*

- *let $\widehat{x}, v$ be defined as Definition 1.1*

- *let $\widehat{z} \in \mathbb{C}^{|\mathcal{F}|}$ be a vector,*

- *let $g(t) = 0$ for any $t \in [0, T]$,*

*then there exists a deterministic procedure* HASHTOBINS$(x, \widehat{z}, H)$ *(see Algorithm 1) which computes* $u \in \mathbb{C}^B$ *with the following guarantees:*

- *Noiseless version: Let* $g(t) \equiv 0$, *for any* $f \in \mathcal{F}$, *the output* $\widehat{u}$ *of* HASHTOBINS$(x, \widehat{z}, H)$ *satisfies*

$$\widehat{u}_{h_{\sigma,b}(f)} = \sum_{f' \in \mathcal{F}} \widehat{G}_{o_{f,\sigma,b}(f')}(v_{f'} - \widehat{z}_{f'})\omega^{a\sigma f'}.$$

- *Noise-only version: Let* $x^*(t) \equiv 0$, *the output* $\widehat{u}$ *of* HASHTOBINS$(g, \mathbf{0}_B, H)$ *satisfies*

$$\|\widehat{u}\|_\infty \leq O(\frac{D}{B} \cdot (C \cdot (\frac{1}{T}\int_0^T |g(t)|^2 \mathrm{d}t + \xi))^{\frac{1}{2}}).$$

- *The algorithm takes* $O(B \log B)$ *samples.*

- *The time complexity of the algorithm is* $O(B \log B)$.

*Proof.* **Part 1: Noiseless version** Let $\widehat{z} := \sum_{f \in \mathcal{F}} \widehat{z}_f \cdot \delta_f(\xi)$, we have

$$
\begin{aligned}
\widehat{u}_{h_{\sigma,b}(f)} &= \widehat{G} * P_{\sigma,a,b}\widehat{(x^* - z)}(h(f)/B) \\
&= \int_{\xi \in \mathbb{R}} \widehat{G}(h_{\sigma,b}(f)/B - \xi)P_{\sigma,a,b}\widehat{(x^* - z)}(\xi)\mathrm{d}\xi \\
&= \int_{\xi \in \mathbb{R}} \widehat{G}(h_{\sigma,b}(f)/B - \xi) \cdot \frac{1}{\sigma}(\widehat{x}(\frac{\xi}{\sigma} + b) - \widehat{z}(\frac{\xi}{\sigma} + b)) \cdot e^{2\pi \mathbf{i}a(\xi + b\sigma)}\mathrm{d}\xi \\
&= \int_{\xi \in \mathbb{R}} \widehat{G}(h_{\sigma,b}(f)/B - \sigma(\xi - b)) \cdot (\widehat{x}(\xi) - \widehat{z}(\xi)) \cdot e^{2\pi \mathbf{i}a\sigma\xi}\mathrm{d}\xi \\
&= \int_{\xi \in \mathbb{R}} \widehat{G}(h_{\sigma,b}(f)/B - \sigma(\xi - b)) \cdot (\sum_{f \in \mathcal{F}}(v_f - \widehat{z}_f) \cdot \delta_f(\xi)) \cdot e^{2\pi \mathbf{i}a\sigma\xi}\mathrm{d}\xi \\
&= \sum_{f \in \mathcal{F}} \widehat{G}(h_{\sigma,b}(f)/B - \sigma(f - b)) \cdot (v_f - \widehat{z}_f) \cdot e^{2\pi \mathbf{i}a\sigma f} \\
&= \sum_{f \in \mathcal{F}} \widehat{G}(-o_{f,\sigma,b}(f)) \cdot (v_f - \widehat{z}_f) \cdot e^{2\pi \mathbf{i}a\sigma f}
\end{aligned}
$$

where the first step is by **Property 2** of Lemma B.4, the 3rd step is by **Property 1** of Lemma B.4, the 4th step is by integral substitution, the 5th step is by the definition of sparse signal (see Definition 1.1), the 6th step is by definition of delta function, the last step is by the definition of hashing functions (see Definition 2.4).

The first part is then proved by the symmetricity of $\widehat{G}$.

**Part 2: Noise-only version**

For any $j \in [BD]$

$$
\begin{aligned}
y_j^2 &= G(j)^2 \cdot x^2(\sigma(t - a)) \\
&\leq O(\frac{1}{B^2}) \cdot x^2(\sigma(t - a)) \\
&\leq O(\frac{1}{B^2}) \cdot (C \cdot \frac{1}{T}\int_0^T |g(t)|^2 \mathrm{d}t + \xi)
\end{aligned}
$$

where the first step is by definition in line 4 of Algorithm 1, the 2nd step is by **Property 5** of Definition 2.8, the 3rd step is by Assumption B.3.

Therefore, we have

$$|y_j| \leq O(\frac{1}{B} \cdot (C \cdot \frac{1}{T}\int_0^T |g(t)|^2 \mathrm{d}t + \xi)^{\frac{1}{2}})$$

We get the result by timing a $D$ (see line 7 of Algorithm 1).

$\square$

## B.4. The Guarantee of median of HASHTOBINS

Combining the previous results, we derive the Guarantee of the median of Deterministic HASHTOBINS.

**Definition B.6** (Choice of $B$). *Let $\alpha$ be some constant to be determined later. Let $k$ be the sparsity of the signal. We define $B$ to be such that*

- $B = \Theta(k)$

- $B$ is a power of $2$

- *we choose the constant in $B = \Theta(k)$ such that the upper bound of Lemma B.7 satisfies:*

$$\Theta(\frac{1}{B}) \leq \frac{1}{\alpha k}$$

**Lemma B.7** (Median of HASHTOBINS outputs). *Under following conditions,*

- *Let $\widehat{x}, v$ be defined as Definition 1.1*

- *Let $\widehat{z} \in \mathbb{C}^{|\mathcal{F}|}$*

- *Let $\widehat{w}_f := v_f - \widehat{z}_f$*

- *We have $\widehat{G}$ being a flat filter (see Definition 2.8)*

- *Let $\{H_r\}_{r \in [d]} = (\sigma_r, a_r, b_r)$ be a sequence of hashing defined in Definition A.6*

- *Let $u_r$ be the output of $\mathrm{HASHTOBINS}(x, \widehat{z}, H_r)$*

*we have*

- *For all $f \in \mathcal{F}$*

$$|\widehat{w}_f - \underset{r \in [d]}{\mathrm{median}}\, \widehat{G}^{-1}_{o_{f,\sigma_r,b_r}(f)}(u_r)_{h_{\sigma_r,b_r}(f)} \cdot \omega^{-a_r \sigma f}|$$
$$\leq \frac{1}{\alpha k} \sum_{f \in \mathcal{F}} |\widehat{w}(f)| + O(\frac{\log k}{k} \cdot (\frac{C}{T} \int_0^T |g(t)|^2 \mathrm{d}t + \xi)^{\frac{1}{2}}) := \mathcal{N}(\widehat{w})$$

*Proof.* Let $u_r^{(noiseless)}$ denote the output of $\mathrm{HASHTOBINS}(x, \widehat{z}, H_r)$ when $g(t) = 0$. Let $u_r^{(noise)}$ denote the output of $\mathrm{HASHTOBINS}(g, \mathbf{0}_{|\mathcal{F}|}, H_r)$.

**Part 1: Guarantee of noiseless HASHTOBINS**

Using Lemma A.21, we know the fifth condition in Lemma B.2 should hold. Then we have for at least $0.8d$ indices $r \in [d]$

$$|\widehat{w}_f - \widehat{G}^{-1}_{o_{f,\sigma_r,b_r}(f)}(u_r)^{(noiseless)}_{h_{\sigma_r,b_r}(f)} \omega^{-a\sigma f}|$$
$$= |\widehat{w}_f - \widehat{G}^{-1}_{o_{f,\sigma_r,b_r}(f)}(m_{H_r}(\widehat{w}(f)))_{h_{\sigma_r,b_r}(f)} \omega^{-a\sigma f}|$$
$$\leq \Theta(\frac{1}{B}) \cdot \|\widehat{w}_{\mathcal{F} \setminus \{f\}}\|_1$$
$$\leq \frac{1}{\alpha k} \cdot \|\widehat{w}_{\mathcal{F} \setminus \{f\}}\|_1 \tag{7}$$

where the first step is by Lemma B.5, the 2nd step uses Lemma B.2, the 3rd is given by choice of $B$ in Definition B.6.

**Part 2: Bounds on output of** HASHTOBINS **with noise function as the input**

Since $\widehat{G}^{-1} \in (0,1)$ by Definition 2.8, combining **Part 2** of Lemma B.5 we have

$$|\operatorname{median} \widehat{G}^{-1}_{o_{f,\sigma_r,b_r}(f)}(u_r)^{(noise)}_{h_{\sigma_r,b_r}(f)}\omega^{-a\sigma f}| \le O(\frac{D}{B} \cdot (\frac{C}{T}\int_0^T |g(t)|^2 \mathrm{d}t + \xi)^{\frac{1}{2}}) \tag{8}$$

**Part 3: Putting them together**

Notice that every operation in HASHTOBINS is linear. Therefore, we have $u_r = u_r^{(noiseless)} + u_r^{(noise)}$.

Then by taking the median of outputs for hash functions in $H_r$, we have,

$$|\widehat{w}(f) - \operatorname{median} \widehat{G}^{-1}_{o_{f,\sigma_r,b_r}(f)}(u_r)_{h_{\sigma_r,b_r}(f)}\omega^{-a\sigma f}|$$

$$\le |\widehat{w}(f) - \operatorname{median} \widehat{G}^{-1}_{o_{f,\sigma_r,b_r}(f)}(u_r)^{(noiseless)}_{h_{\sigma_r,b_r}(f)}\omega^{-a\sigma f}| + |\operatorname{median} \widehat{G}^{-1}_{o_{f,\sigma_r,b_r}(f)}(u_r)^{(noise)}_{h_{\sigma_r,b_r}(f)}\omega^{-a\sigma f}|$$

$$\le \frac{1}{\alpha k}\|\widehat{w}_{\mathcal{F}\setminus\{f\}}\|_1 + |\operatorname{median} \widehat{G}^{-1}_{o_{f,\sigma_r,b_r}(f)}(u_r)^{(noise)}_{h_{\sigma_r,b_r}(f)}\omega^{-a\sigma f}|$$

$$\le \frac{1}{\alpha k}\|\widehat{w}_{\mathcal{F}\setminus\{f\}}\|_1 + O(\frac{D}{B} \cdot (\frac{C}{T}\int_0^T |g(t)|^2 \mathrm{d}t + \xi)^{\frac{1}{2}})$$

where the first step is by triangle inequality, the 2nd step is by Eq. (7), the 3rd step is by Eq. (8).

Hence, we prove the desired result. $\qquad\square$

### B.5. The Guarantee of Super-Linear Time Sparse Recovery

This section presents the super-linear time sparse recovery procedure.

---

**Algorithm 5** Super-Linear time Sparse Recovery for $\widehat{x} - \widehat{z}$

---

1: **procedure** SUPERLINEAR($x \in \mathbb{C}^n$)  $\qquad\qquad\qquad\qquad\qquad\qquad\qquad\qquad\qquad\qquad\qquad$ ▷ Lemma B.8
2: $\quad S \leftarrow \emptyset$
3: $\quad$ **for** $r = 1 \to d$ **do**
4: $\quad\quad u_r \leftarrow$ HASHTOBINS($x, \widehat{z}, (\sigma_r, 0, b_r)$) $\qquad\qquad\qquad\qquad\qquad\qquad\qquad$ ▷ Lemma B.5
5: $\quad$ **end for**
6: $\quad$ **for** $f \in \mathcal{F}$ **do**
7: $\quad\quad \widehat{w}'_f \leftarrow \operatorname{median}_{r \in [d]} \widehat{G}^{-1}_{o_{f,\sigma_r,b_r}(f)}(u_r)_{h_{\sigma_r,b_r}(f)} \cdot \omega^{-a\sigma f}$ $\qquad\qquad\quad$ ▷ Lemma B.7
8: $\quad\quad$ **if** $|\widehat{w}'_f| > \nu/2$ **then**
9: $\quad\quad\quad S \leftarrow S \cup \{f\}$
10: $\quad\quad$ **end if**
11: $\quad$ **end for**
12: $\quad$ **return** $\widehat{w}'_S$
13: **end procedure**

---

The next statement shows core guarantees of Algorithm 5.

**Lemma B.8** (A variation of Lemma 5.2 in Li & Nakos (2020))**.** *If the following conditions hold:*

- *let $\widehat{x}$, $v$ be defined as Definition 1.1*

- *let $\widehat{z} \in \mathbb{C}^{|\mathcal{F}|}$*

- *let $B$ be defined as Definition B.6*

- *let $\widehat{w}_f := v_f - \widehat{z}_f$*

- *Let $\mathcal{N}(\widehat{x})$ be defined as in Lemma B.7*

- *Let $\nu \geq 16\mathcal{N}(\widehat{x})$ be a constant to denote a threshold for heavy index*

*then the output of the Procedure* SUPERLINEAR *(Algorithm 5)* $\widehat{w}'$ *satisfies:*

- $|\widehat{w}_f| \geq (7/16)\nu$ *for all* $f \in \mathrm{supp}(\widehat{w}')$

- $|\widehat{w}_f - \widehat{w}'_f| \leq |\widehat{w}_f|/7$ *for all* $f \in \mathrm{supp}(\widehat{w}')$

- $\{f \in \mathcal{F} : |\widehat{w}_f| \geq \nu\} \subseteq \mathrm{supp}(\widehat{w}')$

*Proof.* **Proof of Part 1**

For any $f \in \mathcal{F}$, we have

$$
\begin{aligned}
|\widehat{w}_f - \widehat{w}'_f| &= |\widehat{w}_f - \underset{r \in [d]}{\mathrm{median}} \, \widehat{G}^{-1}_{o_{f,r}(f)}(u_r)_{h_r(f)} \cdot \omega^{-a\sigma f}| \\
&\leq \mathcal{N} \\
&\leq \frac{\nu}{16}
\end{aligned}
\tag{9}
$$

where the first step is the output of Algorithm 5, the 2nd step is by Lemma B.7, the last step is by the 5th assumption of this lemma.

Then we have,

$$
\begin{aligned}
|\widehat{w}_f| &\geq |\widehat{w}'_f| - |\widehat{w}_f - \widehat{w}'_f| \\
&\geq \nu/2 - |\widehat{w}_f - \widehat{w}'_f| \\
&\geq \nu/2 - \nu/16 = (7/16)\nu
\end{aligned}
\tag{10}
$$

where the first step uses triangle inequality, the 2nd step is by the threshold condition in 8th line of Algorithm 5, the 3rd step is due to Eq. (9).

**Proof of Part 2**

$$
\begin{aligned}
|\widehat{w}_f - \widehat{w}'_f| &\leq \frac{\nu}{16} \\
&\leq \frac{1}{16} \cdot (16/7)|\widehat{w}_f| = |\widehat{w}_f|/7
\end{aligned}
$$

where the first step is given by Eq. (9), the 2nd step is by Eq. (10).

**Proof of Part 3**

$$
\begin{aligned}
|\widehat{w}'_f| &\geq |\widehat{w}_f| - |\widehat{w}_f - \widehat{w}'_f| \\
&\geq \nu - |\widehat{w}_f - \widehat{w}'_f| \\
&\geq \nu - \nu/16 > \nu/2
\end{aligned}
$$

where the 1st step is by triangle inequality, the 2nd step is since $f \in \{f \in \mathcal{F} : |\widehat{w}_f| \geq \nu\}$, the 3rd step is by Eq. (9).

Since $|\widehat{w}'_f|$ is bigger than the threshold, it will be recovered. $\qquad \square$

### B.6. Super-Linear time main algorithm

This section analyzes our main algorithm. First, we present the constraints of the constant parameters.

**Definition B.9** (Constraints of constant parameters)**.** *We list some constraints for constant parameters in the main algorithm.*

- *Part 1 :* $C > 1$

- *Part 2 :* $C(1 - \frac{16}{\alpha}) \geq \frac{16}{\alpha}(1 + \frac{1}{\rho})$

- *Part 3 :* $\frac{7}{16}C \geq \frac{1}{\rho}$

- *Part 4 :* $\gamma \leq 7$

Below is a group of parameters that satisfies the above constraints.

**Definition B.10** (Choice of constant parameters). *We let $C = 2, \rho = 32, \alpha = 32, \gamma = 2$.*

The next lemma shows the guarantee of our super-linear time main algorithm.

**Lemma B.11** ($\ell_\infty$ norm reduction, analogous to Lemma 5.3 in page 13 in Li & Nakos (2020), Lemma 3.8 and Lemma 3.9 in page 10 in Price & Song (2015)). *If following holds*

- *Let $\mu$ be defined as Definition 3.11*

- *Let the SNR $R^*$ (see Definition 3.11) satisfy $R^* \leq (F/\eta)^m$ for some constant parameter $m$*

- *let $\widehat{x}$, $v$ be defined as Definition 1.1*

- *Let $C, \beta, \rho, \gamma$ be some constant to be determined as Definition B.10*

- *Let $r_f^{(t)} := v_f - \widehat{z}_f^{(t)}$*

*For all $0 \leq t \leq T^*$, there is an algorithm (Algorithm 6) outputs a vector $\widehat{w}^{(T^*)} \in \mathbb{C}^n$ which satisfies*

- *Let $I := \{f : |v_f| \geq \frac{\mu}{\rho}\}$, $\widehat{x}(f) = r_f^{(t)}$ for all $f \notin I$*

- *$|r_f^{(t)}| \leq |v_f|$ for all $f$*

- *$\|r_I^{(T)}\|_\infty \leq \nu^{(t)}$*

- *The algorithm takes $O(k^2 \log k \cdot \log(F/\eta))$ samples*

- *The algorithm runs in $O((F/\eta)k \log^2(F/\eta))$ time.*

*Proof.* The proof of the first three claims is by mathematical induction.

**Initial Condition**

The first and second claim clearly holds since $\widehat{z}_f^{(0)} = 0$ for all $f \in \mathcal{F}$ and hence $r_f^{(0)} = \widehat{x}(f)$.

For the 3rd claim, we have

$$\nu^{(0)} = C\mu\gamma^T = C\mu R^* = C\|v\|_\infty > \|v\|_\infty = \|r^{(0)}\|_\infty$$

where the 2nd step is by $T = \log_\gamma R^*$, the 3rd step is by the definition of $R^*$ (see Definition 3.11), the 4th step is by $C > 1$ (**Part 1** of Definition B.9), the last step is by definition of $\|r^{(0)}\|_\infty$.

**Induction step**

Now, we assume the first three claims hold for $t$, we want to prove its correctness for $t + 1$.

For simplicity, we define a notation for the integral of noise on time intervals,

$$J := O(\frac{\log k}{k} \cdot (\frac{C}{T}\int_0^T |g(t)|^2 \mathrm{d}t + \xi)^{\frac{1}{2}})$$

We have

$$\frac{16}{\alpha k} \cdot (\|r^{(t)}\|_1 + J) = \frac{16}{\alpha k} \cdot (\|r_{\mathcal{K}\cap I}^{(t)}\|_1 + \|r_{\mathcal{K}\backslash I}^{(t)}\|_1 + \|r_{\mathcal{F}\backslash\mathcal{K}}^{(t)}\|_1 + J)$$

$$\leq \frac{16}{\alpha k} \cdot (k \cdot \|r_I^{(k)}\|_\infty + \|r_{\mathcal{K}\setminus I}^{(t)}\|_1 + \|r_{\mathcal{F}\setminus\mathcal{K}}^{(t)}\|_1 + J)$$

$$\leq \frac{16}{\alpha k} \cdot (k \cdot C\mu\gamma^{T^*-t} + \|r_{\mathcal{K}\setminus I}^{(t)}\|_1 + \|r_{\mathcal{F}\setminus\mathcal{K}}^{(t)}\|_1 + J)$$

$$\leq \frac{16}{\alpha k} \cdot (k \cdot C\mu\gamma^{T^*-t} + \frac{k\mu}{\rho} + \|r_{\mathcal{F}\setminus\mathcal{K}}^{(t)}\|_1 + J)$$

$$= \frac{16}{\alpha k} \cdot (k \cdot C\mu\gamma^{T^*-t} + \frac{k\mu}{\rho} + k\mu)$$

$$\leq \frac{16}{\alpha k} \cdot (k \cdot C\mu\gamma^{T^*-t} + \frac{\alpha}{16}(1 - \frac{16}{\alpha})Ck\mu)$$

$$\leq \frac{16}{\alpha k} \cdot (k \cdot C\mu\gamma^{T^*-t} + \frac{\alpha}{16}(1 - \frac{16}{\alpha})Ck\mu\gamma^{T^*-t})$$

$$= C\mu\gamma^{T^*-t} := \nu^{(t)}$$

where the first step and 2nd step are trivial calculations, the 3rd step is by the third claim in this lemma and induction hypothesis, the 4th step is by the definition of $I$, the 5th step is by the definition of $\mu$ (see Definition 3.11), the 6th step is derived from **Part 2** of Definition B.9, the 7th step is by $\gamma^{T^*-t} > 1$ and rearrangement.

Therefore, we've proved the fifth condition for Lemma B.8.

Claim 1:

For index $f \in \mathcal{F}\setminus I$, we have

$$\|\widehat{x}(f)\| \leq \frac{\mu}{\rho}$$

$$\leq \frac{7}{16}C\mu$$

$$\leq \frac{7}{16}C\mu\gamma^{T^*-t} := \nu^{(t)}$$

where the first step is by definition of $I$, the 2nd step is by **Part 3** of Definition B.9, the 3rd step is by $\gamma^{T^*-t} > 1$.

By **Part 1** of Lemma B.8, we know that $|\widehat{x}(f)|$ will never be recovered in this procedure.

Claim 2:

Notice that $r_f^{(t)}$ is defined to be $\widehat{x}(f) - \widehat{z}_f^{(t)}$, then we have,

$$|r_f^{(t+1)}| := |\widehat{w}_f - \widehat{w}'_f|$$

$$\leq |\widehat{w}_f|/7$$

$$:= |r_f^{(t)}|/7 \qquad (11)$$

where the first and the third step are by definition of $r$, the 2nd step uses **Part 2** of Lemma B.8

Since we know $|r_f^{(0)}| = |\widehat{x}(f)|$, Claim 2 clearly holds.

Claim 3:

For $f \in I$ such that $|r_f^{(t)}| \leq \nu^{(t+1)}$. Claim 3 is proved by Eq. (11).

Otherwise, we have $|r_f^{(t)}| > \nu^{(t+1)}$, then we have,

$$|r_f^{(t+1)}| \leq |r_f^{(t)}|/7 \leq \nu_f^{(t)}/7 \leq \nu^{(t)}/\gamma := \nu^{(t+1)}$$

where the first step is by Eq. (11), the 2nd step is by induction hypothesis, the 3rd step is by **Part 4** of Definition B.9.

Therefore, Claim 3 holds and we verify the induction step.

**Proof of Sample Complexity.**

The sample complexity of the algorithm is counted as below,

$$\begin{aligned}
\text{Sample Complexity} &= d \cdot \textsc{HashToBins} \\
&= d \cdot O(B \log(B)) \\
&= O(B \log(F/\eta)) \cdot O(\log B) \cdot O(B) \\
&= O(k^2 \log k \cdot \log(F/\eta))
\end{aligned}$$

where the first step is since we have $d$ hashing tuples, the 2nd step is by Lemma B.5, the third step is by choice of $d$ (see Definition A.1), the 4th step is by $B = \Theta(k)$ (see Definition B.6).

Notice that we can reuse the sample, so we only need to count $d$ times of $\textsc{HashToBins}$.

**Proof of Running Time.**

The time complexity of $\textsc{SuperLinear}$ is

$$\begin{aligned}
d \cdot \textsc{HashToBins} + (F/\eta) \cdot \text{Taking Median} &= d \cdot O(k \log k) + F/\eta \cdot \text{Taking Median} \\
&= d \cdot O(k \log k) + (F/\eta) \cdot d \\
&= O(k^2 \log(F/\eta) \log k + (F/\eta)k \log(F/\eta)) \\
&= O((F/\eta)k \log(F/\eta))
\end{aligned}$$

where the 2nd step is because we scan the sequence of hashing when taking the median, the 3rd step is by the choice of $d$, and the 4th step holds for $F/\eta \gg k$.

Hence, the time complexity for the main algorithm is $T^* \cdot O((F/\eta)k \log(F/\eta)) = O((F/\eta)k \log^2(F/\eta))$ since $R^* = O((F/\eta)^m)$ by the first assumption of this lemma.

$\square$

---

**Algorithm 6** Superlinear-time sparse recovery for $\widehat{x}$

---

1: **procedure** $\textsc{Main}(x \in \mathbb{C}^n)$       $\triangleright$ Lemma B.11
2:      $T^* \leftarrow \log_\gamma R^*$
3:      $\widehat{z}^{(0)} \leftarrow \mathbf{0}_{|\mathcal{F}|/\eta}$
4:      $\nu^{(0)} \leftarrow C\mu\gamma^T$
5:      **for** $t = 0 \rightarrow T^* - 1$ **do**
6:          $\widehat{z}^{(t+1)} \leftarrow \widehat{z}^{(t)} + \textsc{SuperLinear}(x, \widehat{z}^{(t)}, \nu^{(t)})$       $\triangleright$ Algorithm 5
7:          $\nu^{(t+1)} \leftarrow \nu^{(t)}/\gamma$
8:      **end for**
9:      **return** $\widehat{z}$
10: **end procedure**

---

### B.7. Main result of the super-linear CFT

This section summarizes the results in previous sections.

**Theorem B.12** (Super-linear Deterministic CFT). *If the following conditions hold*

- *Let $x : [0, T] \rightarrow \mathbb{C}$ has Continuous Fourier Transform $\widehat{x} : [-F, F] \rightarrow \mathbb{C}$*

- *Let the SNR $R^*$ (see Definition 3.11) satisfy $R^* \leq (F/\eta)^m$ for some constant parameter $m$*

- *Let $|\mathcal{F}|, B$ be the powers of 2*

- *Let the noise function $g(t)$ satisfy Definition B.3*

*Then for any vector $x \in \mathbb{C}^n$ satisfying the above assumptions, there is an algorithm that*

- *it finds an $O(k)$-sparse vector $\widehat{x}' \in \mathbb{C}^{|\mathcal{F}|}$*

- *We have $|v_f - \widehat{x}'_f| \leq O(\mathcal{N})$ for all $f \in \mathcal{F}$, where*

$$\mathcal{N} := \frac{1}{\alpha k} \sum_{f \in \mathcal{F}} |v_f| + O\left(\frac{\log k}{k} \cdot \left(\frac{C}{T} \int_0^T |g(t)|^2 \mathrm{d}t + \xi\right)^{\frac{1}{2}}\right)$$

- *The algorithm takes $O(k^2 \log k \cdot \log(F/\eta))$ samples*

- *The algorithm runs in $O((F/\eta)k \log^2(F/\eta))$ time.*

*Proof.* This theorem is a direct result of Lemma B.11.

**Proof of sparsity output**

This is followed by $|I| = O(k)$ and $f \notin I$ is not recovered (**Part 1** of Lemma B.11).

**Proof of Guarantee**

$$\begin{aligned}
||r^{(T^*)}||_\infty &= \max\{||r_I^{(T^*)}||_\infty, ||r_{I^c}^{(T^*)}||_\infty\} \\
&\leq \max\{||\nu^{(T^*)}||_\infty, ||r_{\mathcal{F}\setminus I}^{(T^*)}||_\infty\} \\
&\leq \max\{\nu^{(T^*)}, ||\widehat{x}_{\mathcal{F}\setminus I}^{(T^*)}||_\infty\} \\
&\leq \max\{C\mu, ||\widehat{x}_{\mathcal{F}\setminus I}^{(T^*)}||_\infty\} \\
&\leq \max\{C\mu, (1/\rho)\mu\} = O(\mathcal{N})
\end{aligned}$$

where the 2nd step is by **Part 3** of Lemma B.11, the 3rd step is by bf Part 2 of Lemma B.11, the 4th step is by definition of $\mu$, the 5th step is by definition of $I$.

**Proof of Sample Complexity and Time Complexity**

They are calculated in Lemma B.11. $\qquad\qquad\square$

## C. Sub-linear Time Algorithm

This section provides the missing proofs in Section 3.2, which presents a sub-linear time recovery algorithm.

In Section C.1, we state a number of one-sparse recovery tools from previous work. In Section C.2, we prove the guarantee of sub-linear time sparse recovery.

### C.1. Sub-linear time one-sparse recovery

Now, we introduce a useful lemma that upper bounds the difference in angles.

**Lemma C.1** (Proposition 4.10 in page 11 in Li & Nakos (2020)). *If the following holds,*

- *Let $x, y \in \mathbb{C}$ satisfy $|y| \leq |x|/3$*

*we have*

$$|\arg(x + y) - \arg x| \leq \pi/8$$

We present the one-sparse recovery algorithm for the discrete setting in Li & Nakos (2020) here.

**Lemma C.2** (One-Sparse recovery, Lemma 6.1 in Li & Nakos (2020)). *If the following holds,*

- *$|\mathcal{F}|$ is a power of 2*

- *Let $Q := \{0, 2^0, 2^1, 2^2 \cdots, |\mathcal{F}|/2\}$*

- *Let $x \in \mathbb{C}^{\mathcal{F}}$ with discrete Fourier transform $\widehat{x}$*

- *Let $x_f$ be the $f$-th entry of $x$*

- *Let $\theta_f := \frac{2\pi}{|\mathcal{F}|} f' \mod 2\pi$*

- *Let $\{x_q\}_{q \in Q}$ be a sequence of metric of $x_f$ satisfying*

$$|\arg(x_q) - (\arg \widehat{x}_f + q\theta_f)| \leq \pi/8$$

*There is an algorithm* ONESPARSERECOVERY *(see Algorithm 2) that recover $f$ by $\{x_q\}_{q \in Q}$ in $O(\log F/\eta)$ time.*

### C.2. The Guarantee of Sub-linear time sparse recovery

The next lemma proves the noise in each hashing and filtering bucket is limited, conditioning on bad event does not occur.

**Lemma C.3** (A variation of Lemma 6.2 in page 18 in Li & Nakos (2020)). *If the following conditions hold*

- *Let $H_r = \{\sigma_r, a_r, b_r\}_{r \in d}$ be a sequence of hashing defined in Definition 2.6*

- *We have $\widehat{G}$ being a flat filter with $\epsilon$ buckets and sharpness $\epsilon$ (see Definition 2.8)*

- *Let $\widehat{x}, v$ be defined as Definition 1.1*

- *For all $f \neq f' \in \mathcal{F}$ it holds that,*

$$\sum_{r \in [d]} \widehat{G}^{-1}_{o_{f,\sigma_r,b_r}(f)} \widehat{G}_{o_{f,\sigma_r,b_r}(f')} \leq \frac{\beta}{1-\epsilon}$$

  *where $\beta$ is chosen as Definition A.1.*

*Then for any $f$, at least $0.8d$ indices $r \in [d]$ satisfy*

$$|\sum_{f' \in \mathcal{F}\backslash\{f\}} \widehat{G}_{o_{f,\sigma_r,b_r}(f')} \widehat{x}_f| \leq \Theta(\frac{1}{B}) \cdot \sum_{f' \in \mathcal{F}\backslash\{f\}} |v_{f'}|$$

*Proof.*

$$|\sum_{f' \in \mathcal{F}\backslash\{f\}} \widehat{G}_{o_{f,\sigma_r,b_r}(f')} \widehat{x}_f| = \widehat{G}_{o_{f,\sigma_r,b_r}(f)} \cdot |\widehat{x}_f - \widehat{G}^{-1}_{o_{f,\sigma_r,b_r}(f)} (m_H)_{h(f)} \omega^{-a\sigma f}|$$

$$\leq \widehat{G}_{o_f(f)} \cdot \Theta(\frac{1}{B}) \cdot \sum_{f' \in \mathcal{F}\backslash\{f\}} |v_{f'}|$$

$$\leq \Theta(\frac{1}{B}) \cdot \sum_{f' \in \mathcal{F}\backslash\{f\}} |v_{f'}|$$

where the 1st step is derived from Claim 2.11, the 2nd is from Lemma B.2, the 3rd step holds because $\widehat{G}_{o_{f,\sigma_r,b_r}(f)} < 1$ (see Definition 2.8). □

Now, we show that the one-step guarantees in Lemma B.8 still hold for the sub-linear algorithm.

**Lemma C.4** (A variation of Lemma 6.3 in Li & Nakos (2020)). *If the following conditions hold:*

- *let $\widehat{x}, v$ be defined as Definition 1.1*

- *let $\widehat{z} \in \mathbb{C}^{|\mathcal{F}|}$*

- *let B be defined as Definition B.6*

- *let $\widehat{w}_f := v_f - \widehat{z}_f$*

- *Let $\mathcal{N}(\widehat{w})$ be defined as in Lemma B.7*

- *Let $\nu \geq 16\mathcal{N}(\widehat{w})$ be a constant to denote a threshold for heavy index*

*then the output of the Procedure* SUBRECOVERY *(Algorithm 3) $\widehat{w}'$ satisfies:*

- *For all $f \in \text{supp}(\widehat{w}')$, we have $|\widehat{w}_f| \geq (7/16)\nu$*

- *For all $f \in \text{supp}(\widehat{w}')$, we have $|\widehat{w}_f - \widehat{w}'_f| \leq |\widehat{w}_f|/7$*

- *$\{f \in \mathcal{F} : |\widehat{w}_f| \geq \nu\} \subseteq \text{supp}(\widehat{w}')$*

*Proof.* The proofs of the first and the second statements directly follow Lemma B.8. Now, we prove the third statement. For the $r$-th round, we consider $(\widehat{y}_r)_f := \widehat{G}^{-1}_{o_{f,\sigma_r,b_r}(f)}\widehat{w}_f$ and we use $u_q$ as the metric to recover the $f$ from $(\widehat{y}_r)_f$ (see Line 8 in Algorithm 3). From Lemma C.2, to verify $f$ can be recovered by ONESPARSERECOVERY, we only need to show

$$|\arg(u_q) - \arg(\widehat{G}^{-1}_{o_{f,\sigma_r,b_r}(f)}\widehat{w}_f \cdot \omega^{q\theta_f})| = |\arg(u_q) - (\arg((\widehat{y}_r)_f) + q\theta_f)| \leq \pi/8 \tag{12}$$

By Lemma B.5, we have

$$u_q = \widehat{G}^{-1}_{o_{f,\sigma_r,b_r}(f)}\widehat{w}_f \cdot \omega^{q\theta_f} + \sum_{f' \in \mathcal{F}\setminus\{f\}} \widehat{G}^{-1}_{o_{f,\sigma_r,b_r}(f')}\widehat{w}_{f'} \cdot \omega^{q\theta_{f'}} \pm O\left(\frac{\log k}{k} \cdot \left(\frac{C}{T}\int_0^T |g(t)|^2 \mathrm{d}t + \xi\right)^{\frac{1}{2}}\right) \tag{13}$$

For $f \in \{f \in \mathcal{F} : |\widehat{w}_f| \geq \nu\}$, For at least $8d/10$ repetitions $r \in [d]$, we have

$$|\widehat{G}^{-1}_{o_{f,\sigma_r,b_r}(f)}\widehat{w}_f \cdot \omega^{q\theta_f}|$$
$$\geq (1-\epsilon)\widehat{w}_f$$
$$\geq (1-\epsilon)\nu$$
$$\geq 16(1-\epsilon)\mathcal{N}(\widehat{w})$$
$$= 16(1-\epsilon)\left(\frac{1}{\alpha k}\sum_{f \in \mathcal{F}}|\widehat{w}(f)| + O\left(\frac{\log k}{k} \cdot \left(\frac{C}{T}\int_0^T |g(t)|^2\mathrm{d}t + \xi\right)^{\frac{1}{2}}\right)\right)$$
$$\geq 16(1-\epsilon)\left(\frac{\Theta(B)}{\alpha k}|\sum_{f' \in \mathcal{F}\setminus\{f\}}\widehat{G}_{o_{f,\sigma_r,b_r}(f')}\widehat{x}_f| + O\left(\frac{\log k}{k} \cdot \left(\frac{C}{T}\int_0^T |g(t)|^2\mathrm{d}t + \xi\right)^{\frac{1}{2}}\right)\right)$$
$$\geq 16(1-\epsilon)\left(\frac{\Theta(k)}{\alpha k}|\sum_{f' \in \mathcal{F}\setminus\{f\}}\widehat{G}_{o_{f,\sigma_r,b_r}(f')}\widehat{x}_f| + O\left(\frac{\log k}{k} \cdot \left(\frac{C}{T}\int_0^T |g(t)|^2\mathrm{d}t + \xi\right)^{\frac{1}{2}}\right)\right)$$
$$\geq 3\left(|\sum_{f' \in \mathcal{F}\setminus\{f\}}\widehat{G}_{o_{f,\sigma_r,b_r}(f')}\widehat{x}_f| + O\left(\frac{\log k}{k} \cdot \left(\frac{C}{T}\int_0^T |g(t)|^2\mathrm{d}t + \xi\right)^{\frac{1}{2}}\right)\right)$$
$$\geq 3|u_q - \widehat{G}^{-1}_{o_{f,\sigma_r,b_r}(f)}\widehat{w}_f \cdot \omega^{q\theta_f}|$$

where the first step is by $\widehat{G}^{-1} \geq 1 - \epsilon$, the 2nd step is by $f \in \{f \in \mathcal{F} : |\widehat{w}_f| \geq \nu\}$, the 3rd step is by the definition of $\mathcal{N}(\widehat{w})$, the 4th step uses Lemma C.3, the 5th step is by $B = \Theta(k)$, the 6th step is by choosing a proper $\alpha$ according to the constant hided in $\Theta(k)$, the last step is by Eq. (13).

Therefore, using Lemma C.1 with $x = \widehat{G}^{-1}_{o_{f,\sigma_r,b_r}(f)}\widehat{w}_f \cdot \omega^{q\theta_f}$ and $x + y = u_q$ proves Eq. (12).

For at least $8d/10$ repetitions $r \in [d]$, the above argument holds and we can recover the wanted frequency. Therefore, we proved the desired result. □

