# OpenReview forum: "Deterministic Sparse Fourier Transform for Continuous Signals with Frequency Gap"
_ICML.cc/2025/Conference — ICML 2025 poster_

### Official Review · Reviewer_gLST · 2025-03-14

**Overall Recommendation:** 4

**Summary:**

The paper introduces the first deterministic algorithm for computing the sparse Fourier transform (SFT) of continuous signals that have a minimum frequency gap. In contrast to earlier approaches that relied on randomness, the authors develop a method that deterministically recovers a k‑sparse signal (i.e., one with only k significant frequencies) using far fewer samples than traditional FFT methods. Their algorithm uses a de‑randomized hashing scheme combined with specialized filtering functions to isolate individual frequency components even in the presence of noise. A novel (C, ξ)-noise model (defined below) is employed to guarantee robust recovery under an ℓ₁/ℓ₂ mixed norm error bound. Overall, the method achieves sublinear sample complexity and runtime—specifically O(k² polylog(FT/η))—making it an optimal deterministic solution for continuous sparse Fourier transforms when frequencies are separated by a gap.

A noise function g(t) defined on the interval [0, T] is considered (C, ξ)-noise if its maximum squared magnitude over [0, T] is upper-bounded by a constant multiple of its average energy plus an additive term. Formally, this means that

  max₍ₜ∈[0,T]₎ |g(t)|² ≤ C · (1/T · ∫₀ᵀ |g(t)|² dt) + ξ,

where C is a fixed constant and ξ is a parameter that depends on the specific characteristics of g(t). This condition ensures that even if g(t) has occasional high peaks, the overall noise level remains controlled relative to its average energy.

## Update After Rebuttal:

I think the authors have responded to my review sufficiently and I will keep my score.

**Claims And Evidence:**

Yes, the claims made in the submission are supported by proofs.

**Essential References Not Discussed:**

I think the references are adequate.

**Experimental Designs Or Analyses:**

N/A (this is a theoretical paper)

**Methods And Evaluation Criteria:**

N/A (this is a theoretical paper)

**Other Comments Or Suggestions:**

None

**Other Strengths And Weaknesses:**

The paper offers an original contribution by extending deterministic sparse Fourier techniques from the discrete to the continuous setting. Its strengths include:

• A creative combination of ideas from discrete SFT (e.g., Li & Nakos, Hassanieh et al., Indyk & Kapralov) with continuous signal analysis, leading to a deterministic algorithm that avoids randomness.

• Rigorous theoretical analysis with optimal sublinear sample and runtime guarantees under a (C, ξ)-noise model.

• Clear advancement in theory by removing randomness, which is significant for applications where deterministic performance is critical.

I don't have any significant weaknesses to mention, other than perhaps checking the applicability of this algorithm in a practical context.

**Questions For Authors:**

None

**Relation To Broader Scientific Literature:**

The paper extends deterministic sparse Fourier techniques from the discrete setting—drawing on work by Li & Nakos (2020), Hassanieh et al. (2012), and Indyk & Kapralov (2014)—to continuous signals by introducing a deterministic hashing scheme and a (C, ξ)-noise model. In contrast to previous randomized methods (Price & Song, 2015; Chen et al., 2016; Jin et al., 2023) that rely on random time sampling for sublinear recovery, this work adapts tools such as hash-to-bins and one-sparse recovery to achieve optimal deterministic recovery guarantees.

**Theoretical Claims:**

I read the first 9 pages, and the arguments there seem fine.

---

> ### Author Rebuttal · Authors · 2025-03-31
>
> We thank the reviewer for the valuable comment and we greatly appreciate the reviewer's recognition of our contributions. We believe that our work represents a substantial advancement in the theoretical understanding of Fourier transforms. We also appreciate the reviewer’s perspective regarding the practical applicability of our framework. While our main focus has been on establishing the theoretical foundations, we share the reviewer’s interest in examining real‐world deployments and it would be interesting to see our deterministic method can be evaluated under a variety of practical conditions. Please let us know if you have any further comments regarding our work. Thank you again for your positive feedback.

---

### Official Review · Reviewer_k8uZ · 2025-03-19

**Overall Recommendation:** 3

**Summary:**

This paper introduces a deterministic sublinear-time algorithm for recovering sparse continuous signals with frequency gaps, which addresses a critical gap in prior research by random approaches. The proposed method achieves optimal recovery guarantees in the presence of arbitrary noise.

**Claims And Evidence:**

Yes, the claims are strictly proven.

**Essential References Not Discussed:**

No.

**Experimental Designs Or Analyses:**

The work lacks experiments.

**Methods And Evaluation Criteria:**

The proposed methods are theoretically novel and address a critical gap in continuous sparse recovery. However, the evaluation criteria rely excessively on theory, lacking empirical validation and baseline comparisons, which limits practical credibility.

**Other Comments Or Suggestions:**

No.

**Other Strengths And Weaknesses:**

Strengths:
- Deterministic algorithm for continuous sparse Fourier transform. The paper introduces a deterministic *sublinear-time* algorithm for recovering sparse continuous signals with frequency gaps.
- Tight theoretical guarantees. Theoretical guarantees ensure stable recovery even with arbitrary noise, which addresses a key challenge in continuous signal processing.



Weaknesses:
- Lack of empirical validation. The paper provides no experiments to validate the results.
- No comparison to on-grid compressed sensing. The work does not compare its approach to on-grid compressed sensing methods, which discretize the frequency domain and achieve good sample complexity under the same setting.
-  Grid Assumption is too strong. Real-world signals often exhibit off-grid frequencies, and the algorithm’s performance would degrade significantly in such cases, limiting its applicability.

## update after rebuttal
As the empirical validation is yet to be incorporated, I'll maintain my score as it is.

**Questions For Authors:**

1. How does the algorithm perform on synthetic signals with varying noise levels?
2. Is there a mechanism to handle the grid mismatch?
3. Could you compare the experimental performance of  on-grid compressed sensing and your algorithm by conducting experiments?  For fairness,  they can use the same samples.

**Relation To Broader Scientific Literature:**

The key contributions of this paper address critical gaps in the sparse Fourier transform (SFT) literature, particularly in the context of continuous signals with frequency gaps. Its deterministic guarantees and noise resilience position it as a critical step toward reliable, real-world sparse signal processing systems.

**Theoretical Claims:**

Part of them. And I think they are right.

---

> ### Author Rebuttal · Authors · 2025-03-31
>
> We thank the reviewer for the recognition of our theoretical and algorithmic contributions.
> ### W1, Q1 and Q3: Lack of empirical validation
> Thank you for pointing this out. We acknowledge that empirical validation would be beneficial, but our focus in this submission was on establishing theoretical guarantees and sublinear‐time recovery. Like many other works in this line of research, we have chosen to emphasize these theoretical foundations, focusing on the sublinear‐time property and performance guarantees. Such a focus often entails specialized data structures and asymptotic analyses rather than extensive empirical testing.
>
> ### W2: Comparison of on-grid compressed sensing
> We thank the reviewer for this valuable point. We emphasize that our setting is fundamentally different from standard on-grid compressed sensing methods, which assume discretized frequency domains. Our problem formulation assumes continuous-time signals with non-zero frequency gaps, where naive discretization would suffer from sparsity blow-up and noise sensitivity.
>
> ### W3 and Q2: Grid assumption
> Thank you for the thoughtful feedback. We acknowledge this concern and would like to clarify. Our current algorithm assumes frequencies lie on an equispaced grid with a known frequency gap, as is standard in prior deterministic SFT work. This assumption allows us to design deterministic hashing and filtering strategies with provable guarantees. Extending deterministic sparse recovery to off-grid frequencies is more challenging. We believe one potential way is to refine the grid resolution so that frequencies are only “mildly off-grid,” at the cost of increased sample complexity.
>
> We are grateful for the reviewer’s insightful comments. Thank you for your time and valuable feedback!

---

### Official Review · Reviewer_FmMR · 2025-03-24

**Overall Recommendation:** 4

**Summary:**

This paper adapts Li and Nakos (2020)'s deterministic sparse Fourier transform (SFT) algorithm to the continuous-time setting described by Price and Song (2015) (who had proposed a randomized algorithm), showing that an efficient deterministic method exists in this regime as well. The proposed algorithm has $O(k^2\log k \log^2(F/\eta))$ sample complexity and $O(k^2 \log k \log^3(F/\eta))$ time complexity, where $k$ is the sparsity, $F$ is the bound on possible frequencies, and $\eta$ is the gap between possible frequencies. The algorithm is sublinear with respect to the frequency domain size $\Theta(F/\eta)$, and thus is an improvement over non-sparse algorithms.  The class of $(C, \xi)$-noise functions is defined in Definition 3.3 so that the algorithm can work in the continuous setting. Proofs of the recovery guarantees and sample/time complexities of the algorithm (as summarized in Theorem 3.11) are provided in the appendices.

**Claims And Evidence:**

The main claims of this paper, the correctness and sublinear complexity of the proposed deterministic continuous SFT algorithm, is well substantiated with proofs in the appendix. The required assumptions (e.g., on the noise) are laid out clearly. However, the claim that the algorithm achieves optimal recovery guarantees (see, e.g., the last sentence of the abstract or the second last sentence of the conclusion) lacks justification. The authors should clarify this and explain how the recovery guarantee achieved by the algorithm is optimal.

**Essential References Not Discussed:**

The relevant literature on continuous and/or sparse Fourier transforms seems to be adequately represented in this paper.

**Experimental Designs Or Analyses:**

N/A (No experiments.)

**Methods And Evaluation Criteria:**

The problem definition, recovery guarantees, and use of sample/runtime complexity to evaluate the efficiency of the algorithm are standard in the literature.

**Other Comments Or Suggestions:**

1. In the introduction, a brief discussion on the different kinds of continuous settings studied in the context of SFTs might be beneficial to give readers additional context. For example, although Boufounos et al. (2012) is mentioned in the introduction as a paper that studies SFTs in the continuous setting, it focuses on the discrete-time and continuous-frequency setting, while this paper studies the continuous-time and discrete-frequency setting.
2. There are many minor typographical and formatting errors, which can sometimes impede understanding. For example, the definition of $\omega$ to be $e^{-2 \pi \mathbf{i}}=1$ in the second last line of the right column of page 2 does not make a lot of sense. It seems substituting $e^{-2 \pi \mathbf{i}}$ in for every instance of $\omega$ will fix this issue (e.g., replace $\omega^{t\sigma b}$ with $e^{-2\pi\mathbf{i} t \sigma b}$). In addition, in line 233 of the left column of page 5, "if $o_{f,\sigma,b}(f)$ is big and $o_{f,\sigma,b}(f')$ is small" should probably be either "if $o\_{f,\sigma,b}(f)$ is small and $o\_{f,\sigma,b}(f')$ is big" or "if $\hat{G}\_{o\_{f,\sigma,b}(f)}$ is big and $\hat{G}\_{o\_{f,\sigma,b}(f')}$ is small".

**Other Strengths And Weaknesses:**

1. This paper assumes that the reader is familiar with the general techniques used for sparse Fourier transforms and often defines variables or functions without explaining their purpose. This makes this paper very challenging to read for readers who are not familiar with the subject matter. In addition, this paper borrows many definitions from Li and Nakos (2020), but leaves out the explanations that accompanied them in the original paper. For example, the role of the pessimistic estimator $h\_r (f, f', \sigma\_1, b\_1, \dots,\sigma\_r,b\_r)$ defined in line 262 of the left column of page 5 is not explained, making it very hard to understand for readers who have not read Li and Nakos (2020).
2. It is unclear how useful the setting considered in this paper is. Although the setting allows for continuous-time signals, this is offset by the requirement that all active frequencies be a bounded discrete multiple of some constant $\eta$, which seems rather limiting. Furthermore, the $(C, \xi)$-noise model defined in this paper seems to exclude some common types of noise such as white Gaussian noise.

**Questions For Authors:**

1. The paper claims that the proposed algorithm achieves optimal recovery guarantees. Could the authors explain what it means for a recovery guarantee to be optimal, and how the proposed algorithm is optimal in that sense? (see section "Claims and Evidence")
2. Could the authors provide concrete examples of $(C, \xi)$-noise functions, preferably with exact values for $C$ and $\xi$? Providing examples of commonly encountered $(C, \xi)$-noise functions will help justify the definition. (see section "Other Strengths and Weaknesses", point 2)
3. Could the authors provide potential applications of the continuous-time, discrete-frequency SFT? It is true that this is not the first paper that studies this setting, but some examples of real-world uses will help motivate the problem that this paper is solving. Price and Song (2015) did mention piano tuning as an example, but it seemed rather contrived. (see section "Other Strengths and Weaknesses", point 2)

## Update after Rebuttal
The authors' response was reasonable and addressed most of my concerns. I believe that the final paper will be much stronger if the authors incorporate this discussion in the final version (especially their responses to questions 1-3) and make all the fixes that they promise. Taking this into account, I have increased the overall rating from 3 to 4.

**Relation To Broader Scientific Literature:**

This paper combines the methodology of Li and Nakos (2020) and the setting of Price and Song (2015). Although neither the idea nor the setting is new, the combination seems to be novel. The proposed algorithm is similar to that of Li and Nakos (2020), but has nontrivial differences stemming from the difference in setting.

**Theoretical Claims:**

The proofs of algorithm correctness and complexity (presented in the appendices) generally follow those in Li and Nakos (2020). Although the high-level structure of the proof is similar, significant changes are required due to the continuous-time setting. Although I did not check every detail, it seems that all the necessary adjustments for the proof to go through have been made.

---

> ### Author Rebuttal · Authors · 2025-03-31
>
> We thank the reviewer for the valuable feedback and for recognizing the novelty, significance, and technical contributions of our work.
> ### W1: Self-contained Explanation
> We will include clearer explanations of fundamental Sparse Fourier Transform (SFT) methods (such as hashing, filtering, and convolution) to ensure the paper is easy to understand and complete on its own. For instance, we will clarify the definition of $h_r$ in our revision, stating explicitly: "In the derandomization phase, each function $h_r$ serves as a pessimistic estimator, tracking the probability of undesirable events (such as hash collisions) given the first $r$ selected hash functions." We will carefully define these key concepts throughout the paper.
>
> ### W2 and Q2: $(C,\xi)$-noise model
> We introduced the $(C,\xi)$-noise model to guarantee that if the noise function has bounded pointwise deviation relative to its global energy, a deterministic sampling pattern can isolate true signal components from the noise. White Gaussian noise typically has unbounded probability tails, which can lead to arbitrarily large amplitudes at deterministic sample points and hence one cannot “average out” them in a purely deterministic algorithm. However, our model does not strictly exclude Gaussian noise if it is truncated. Suppose that $g(t)$ is i.i.d. over time t. If we consider $g(t) \sim \mathcal{N}(0,\sigma^2)$ condition on $|g(t)| \leq M$, then it is a $(C,\xi)$-noise with $C=1$ and any $\xi > 0$. In fact, any uniformly bounded process (including a truncated normal) is trivially a $(C,\xi)$-noise. Another example is that $g$ is a polynomial function, e.g. $g(t) = t^c$ for some $c > 0$. Then $g$ is a $(C,\xi)$-noise with $C = 2c+1$ and $\xi = 0$.
> ### Q1: Optimal recovery guarantee
> By “optimal,” we mean that up to polylogarithmic factors in $k$. Our algorithm achieves both optimal sample complexity and runtime. This is because the lower bound for sample complexity is $\Omega(k^2 + k \log k)$, as established in [1]. Consequently, the lower bound for runtime is also $\Omega(k^2 + k \log k) $.
> ### Q3: Applications and real-word examples
> SFT problems arise whenever signals are (approximately) dominated by a small number of frequencies. While classical FFT requires sample/time complexity on the order of the entire band-limit $F$, SFT leverages sparsity $k \ll F$ to reduce complexity. Many continuous-time signals in scientific and engineering contexts are indeed “nearly sparse,” with only a few truly significant frequency components amidst a large potential range of smaller, negligible ones. For example, in many radar systems, the received signal consists of only a few dominant sinusoidal components [2], each corresponding to a strong reflection path. While the overall bandwidth FFF may be large, the effective sparsity (the number of meaningful reflections kkk) is typically quite small. Another example is machinery vibration analysis [3]. Machinery vibrations are inherently continuous signals that often exhibit only a small handful of resonant frequencies—each well spaced from the others. This distinct frequency gap naturally suits a continuous sparse Fourier approach for robust fault detection and monitoring.
>
> [1] Ganguly, Sumit. "Lower bounds on frequency estimation of data streams." Computer Science–Theory and Applications: Third International Computer Science Symposium. 2008
>
> [2] Austin, Christian D., Emre Ertin, and Randolph L. Moses. "Sparse signal methods for 3-D radar imaging." IEEE Journal of Selected Topics in Signal Processing. 2010
>
> [3] Ding, Chuancang, Ming Zhao, and Jing Lin. "Sparse feature extraction based on periodical convolutional sparse representation for fault detection of rotating machinery." Measurement Science and Technology. 2020
>
> We also thank the reviewer for the editorial comments. We appreciate these suggestions and will refine the final draft accordingly. Thank you for your time and valuable feedback!

---

### Decision · Program_Chairs · 2025-05-01

**Decision:**

Accept (poster)

**Comment:**

The paper proposes the first sublinear deterministic algorithm for continuous-time sparse further transform with guaranteed sample and time complexities which appear to be optimal. Naturally, the paper adopts tools from existing papers, but as the reviewers point out the adaptation seems nontrivial, and techniques developed in doing so could be of interest to the theoretical ML community.

The authors' rebuttal further addressed most of reviewer's FmMR initial concerns, particularly regarding the inclusion of necessary background, examples, and definitions to help the reader understand the setting and techniques. Further, the reviewers unanimously agree that the paper's contribution is solid and recommend acceptance.

While the paper is outside the expertise of AC, upon further examination, the AC further agrees with the reviewers and believe the paper is a solid theoretical contribution to the ICML community and clearly exceeds the acceptance bar.